# Cryo-EM structure of human PAPP-A2 and mechanism of substrate recognition

Janani Sridar[1,3], Amirhossein Mafi[1,3], Russell A. Judge [2], Jun Xu[1], Kailyn A. Kong[1], John C. K. Wang [1],
Vincent S. Stoll[2], Georgios Koukos [1], Reyna J. Simon[1], Dan Eaton [1], Matthew Bratkowski [1,4✉] &
Qi Hao [1,4✉]

Pregnancy-Associated Plasma Protein A isoforms, PAPP-A and PAPP-A2, are metalloproteases that cleave insulin-like growth factor binding proteins (IGFBPs) to modulate insulin-like growth factor signaling. The structures of homodimeric PAPP-A in complex with IGFBP5 anchor peptide, and inhibitor proteins STC2 and proMBP have been recently reported. Here, we present the single-particle cryo-EM structure of the monomeric, N-terminal LG, MP, and the M1 domains (with the exception of LNR1/2) of human PAPP-A2 to 3.13 Å resolution. Our structure together with functional studies provides insight into a previously reported patient mutation that inactivates PAPP-A2 in a distal region of the protein. Using a combinational approach, we suggest that PAPP-A2 recognizes IGFBP5 in a similar manner as PAPP-A and show that PAPP-A2 cleaves IGFBP5 less efficiently due to differences in the M2 domain. Overall, our studies characterize the cleavage mechanism of IGFBP5 by PAPP-A2 and shed light onto key differences with its paralog PAPP-A.

[1] Calico Life Sciences LLC, South San Francisco, CA 94080, USA. [2] AbbVie, 1 North Waukegan Rd, North Chicago, IL 60064, USA. [3]These authors contributed equally: Janani Sridar, Amirhossein Mafi. [4]These authors jointly supervised this work: Matthew Bratkowski, Qi Hao. ✉email: mbratkowski@calicolabs.com; qhao@calicolabs.com

Growth hormone stimulates the liver to produce insulin-like growth factor-1 (IGF-1) while the related protein insulin-like growth factor 2 is produced independent of growth factor stimulation[1, 2]. Insulin-like growth factors (IGFs) are secreted in the bloodstream and their binding to IGF-1 receptor results in signaling cascades promoting growth in most tissues[1]. IGFs are bound by six IGF binding proteins (IGFBPs, IGFBP1-6) that extend the half-lives of IGFs and prevent their binding to IGF-1 receptor[3,4]. The proteases PAPP-A and PAPP-A2 cleave specific IGFBPs to release IGFs and stimulate growth factor signaling[5,6].

PAPP-A and PAPP-A2 are secreted, glycosylated proteins containing pro-peptide sequences that are processed to mature forms of the proteins consisting of 1547 and 1557 amino acids, respectively[5–7]. PAPP-A2 shares 46% sequence conservation with PAPP-A (Supplementary Fig. 1, with mature numbering for both proteins used in the text). Both PAPP-A2 and PAPP-A contain an N-terminal Laminin-G (LG) domain and a catalytic metalloprotease (MP) domain that itself contains two Lin12-Notch repeats (LNR1 and LNR2)[8,9]. Both proteins also contain central M1 and M2 domains, five complement control protein domains (CCP1-5), and a C-terminal LNR3 domain (Fig. 1a)[5,10]. Both PAPP-A and PAPP-A2 are inhibited by stanniocalcin-1 and stanniocalcin-2 (STC1 and STC2)[11,12]. The majority of PAPP-A is found in an inhibited form in a 2:2 stoichiometric, covalent complex with the pro-protein form of eosinophil major basic protein (proMBP) in serum during pregnancy[13], but PAPP-A2 does not form a covalent complex with proMBP in serum[14]. Also, PAPP-A binds to cell surface glycosaminoglycans via its CCP3 and CCP4 domains, but PAPP-A2 does not[15]. PAPP-A was first experimentally shown to form a *trans*-homodimer[16] and was later visualized as a dimer by cryo-EM reconstruction in the apo form and when bound to IGFBP5[17], and when bound to inhibitory proteins STC2 and proMBP[18,19]. Unlike PAPP-A, PAPP-A2 was originally shown to not form a covalent dimer by non-reducing polyacrylamide gel electrophoresis (PAGE)[6], yet the majority of PAPP-A2 was shown to run at a similar migration as dimeric PAPP-A on native PAGE[16]. Purified recombinant PAPP-A2, however, was shown to exist as a monomer in solution as analyzed by size exclusion chromatography multi-angle light scattering[17].

PAPP-A and PAPP-A2 are members of the pappalysin family within the broad metzincin protease superfamily whose members feature a zinc-bound active site and a characteristic methionine turn residue[20, 21]. PAPP-A specifically cleaves IGFBP2, IGFBP4, and IGFBP-5, and its cleavage of IGFBP2 and IGFBP4 is IGF-dependent, while PAPP-A2 cleaves IGFBP3 and IGFBP5 in an IGF independent manner[6,7,22–24]. Protein homologs of PAPP-A and PAPP-A2 have also been studied extensively in zebrafish and shown to feature different substrate specificities[25–27]. There are several key determinants of PAPP-A required for cleavage of IGFBP4 but not for IGFBP5 including (1) LNR center formation[9], (2) an interaction between the LG and CCP2 domains[17], and (3) *trans*-dimerization[9,16,17]. IGFBP5 is the only substrate shared between PAPP-A and PAPP-A2[6,23]. We previously reported the cryo-EM structure of PAPP-A bound to a region of IGFBP5 (hereafter called PAPP-A$_{BP5}$) consisting of amino acids 119-143 (hereafter referred to as the anchor peptide)[17]. However, the mechanism of PAPP-A2 substrate recognition remains unknown.

Emerging evidence suggests that PAPP-A2 is important for growth and is non-redundant with PAPP-A[28]. PAPP-A2 knockout mice show growth defects and an increase of total IGF-1 but a decrease in free IGF-1[29,30]. In agreement with mouse studies, PAPP-A2 plays a critical role in postnatal human growth. Mutations result in elevated IGFBP3, IGFBP5, and high total IGF-1 but decreased free IGF-1 as well as insulin resistance and low bone mineral density[31–33]. Frameshift, nonsense, and the A1033V point mutation in PAPP-A2 (referred to as A799V hereafter based on the mature sequence numbering) were shown to result in short stature in patients and abolish cleavage of IGFBP3 and IGFBP5 by PAPP-A2[28,34]. Interestingly, the A799V point mutation site is far from the proteolytic site of PAPP-A2 so the mechanism of how it abolishes PAPP-A2 proteolytic activity is unclear, but the mutant protein was previously reported to have lower expression compared to wildtype protein and was partially cleaved[34].

In this report we use a combinational approach including cryogenic electron microscopy (cryo-EM) structural determination, functional activity assays, machine learning (ML) modeling, and molecular dynamic (MD) simulations to study the mechanism of IGFBP5 cleavage by PAPP-A2. We demonstrate both structural and function similarities and key differences between PAPP-A2 and PAPP-A. Our structure and functional assays also provide insight into PAPP-A2 inactivation by the previously reported A799V patient mutation. These results together uncover specific roles of PAPP-A2 and provide insight into possible interventions for preventing growth defects.

## Results

**Cryo-EM structure of PAPP-A2 and functional analysis reveal a similar IGFBP5 cleavage mechanism as PAPP-A.** We pursued a cryo-EM reconstruction of PAPP-A2 to gain experimental structural insight into its mode of substrate recognition. We used the reported catalytically inactivating E500Q mutant[6] to prevent substrate cleavage, and reconstituted PAPP-A2 with IGFBP5 in attempts to resolve a substrate-bound complex. However, no IGFBP5 density was observed in the structure, suggesting substrate dissociation during the preparation process, and thus we treated the structure as apo PAPP-A2. The structure was reconstructed to 3.13 Å resolution (Fig. 1b, Supplementary Table 1, Supplementary Data 1, and Supplementary Fig. 2a–e), which allowed us to build a model corresponding to residues 25-926, except for residues 373 – 420 in the LNR1 and LNR2 regions that featured ambiguous map density. Therefore, the final structure contains the N-terminal LG, MP, and the M1 domains of PAPP-A2. No density was present for the M2 domain, CCP1-5 domains, or the LNR3 domain, suggesting that these regions are flexible. PAPP-A2 is glycosylated and we observed clear density corresponding to stabilized glycans bound to residues N328 and N445 that we modeled as N-acetylglucosamine (Supplementary Fig. 2f, yellow sticks). We also observed calcium bound to two sites (Supplementary Fig. 2g), analogous to calcium binding sites found in PAPP-A[18,19].

PAPP-A2 is a monomer by cryo-EM reconstruction (Fig. 1b, Supplementary Fig. 2a, b). We confirmed the monomeric nature of recombinant PAPP-A2 by reducing and non-reducing sodium dodecyl sulfate (SDS)-PAGE (Supplementary Fig. 3a) indicating that PAPP-A2 does not form a covalent dimer and in agreement with prior results[6]. PAPP-A2 also eluted as a monomer when run on a size-exclusion chromatography column compared to dimeric PAPP-A (Supplementary Fig. 3b), in agreement with the molecular weight measured previously by size exclusion chromatography multi-angle light scattering[17]. Native PAGE analysis of our recombinant PAPP-A2 also indicated that it was predominantly monomeric (Supplementary Fig. 3c), in contrast to a prior result obtained from Western blotting of PAPP-A2 overexpression cell culture media[16]. We therefore conclude that PAPP-A2 is monomeric in contrast to the dimeric PAPP-A.

The observed PAPP-A2 domains align with the N-terminal domains of one monomer of the PAPP-A$_{BP5}$ structure with a root mean squared deviation (RMSD) score of 2.0 Å (Fig. 1c). The

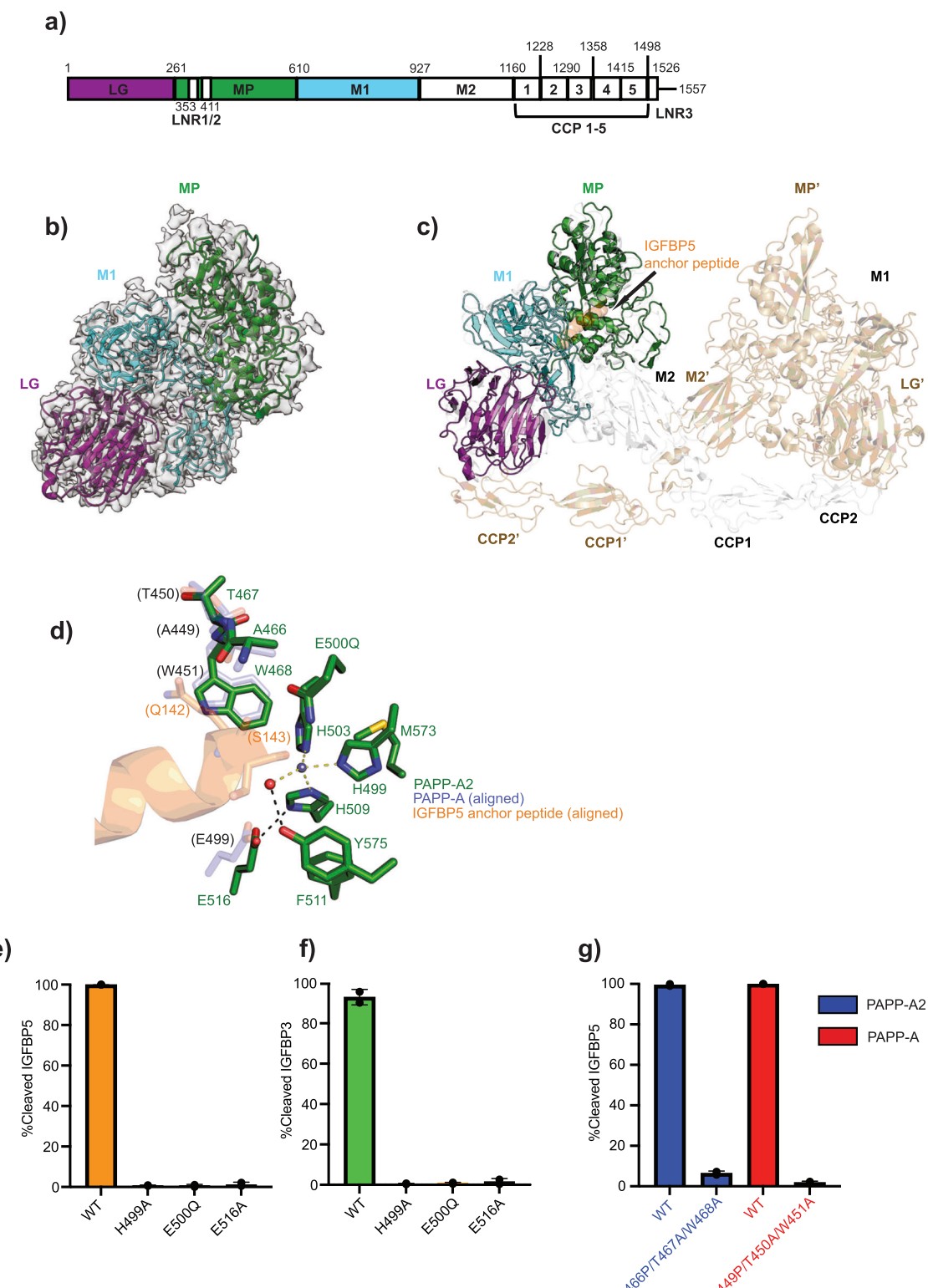

active sites of PAPP-A and PAPP-A2 overlay with an RMSD of 0.35 Å (Fig. 1d). PAPP-A2 features a characteristic metzincin HEXXHXXGXXH motif[21] where zinc is coordinated by H499, H503, and H509, and the catalytic E500 (mutated to Q in our structure) is 3.2 Å away from the histidine cluster (Fig. 1d, with map density shown in Supplementary Fig. 2h). Residue H509 is additionally stabilized by hydrogen bonding to E516 and π-π stacking with F511. Interestingly, we observed the catalytic water that is coordinated by zinc and hydrogen bonds to Y575 (Fig. 1d).

The active site also contains the canonical methionine turn residue M573. Compared with wild type (WT) PAPP-A2, the H499A and E516A mutants abolish cleavage for both IGFBP5 and IGFBP3 like the previously reported E500Q mutant[6], in agreement with analogous results with PAPP-A[20,35] (Fig. 1e, Supplementary Fig. 4a, b; and Fig. 1f, Supplementary Fig. 4a, and Fig. 4c, respectively, and Supplementary Data 2). In the PAPP-A$_{BP5}$ structure, the anchor peptide binds in a groove created by the M1 and M2 domains extending into the cleavage site in the

**Fig. 1 Cryo-EM reconstruction of PAPP-A2 at 3.13 Å resolution. a** Schematic domain organization of PAPP-A2. Numbering shown here and discussed in the text refers to mature PAPP-A2 after removal of the signal secretion and pro-peptide sequences. Domains corresponding to observed density are color-coded while unresolved domains are shown in white. **b** Cryo-EM map density of PAPP-A2 monomer with domains colored according to Fig. 1a. **c** Alignment of PAPP-A2 (colored domains) with one copy of PAPP-A$_{BP5}$ (PDB 7ufg). PAPP-A is shown in transparent gray and olive cartoon and one copy of the anchor peptide is shown as orange cartoon. **d** PAPP-A2 active site. PAPP-A2 MP domain residues are shown as green sticks. Zinc and water are in gray and red spheres, respectively. Zinc coordination bonds are shown as yellow dashes and hydrogen bonds are black dashes. Aligned PAPP-A residues (from PDB 7ufg) are shown as plum sticks and IGFBP5 anchor peptide is shown as transparent orange cartoon with sticks. Residues for both PAPP-A and IGFBP5 are labeled in parentheses to differentiate them from PAPP-A2 residues. **e** Cleavage assays for wildtype and active site PAPP-A2 mutants using IGFBP5 as the substrate. For assays containing IGFBP5 as the substrate shown here and in the remainder of the report, unless indicated otherwise, a concentration of 500 nM was used and reactions were incubated for 4 h at 37 °C (please see METHODS section for further details). Data shown in graphs here and in the rest of the report, unless indicated otherwise, represent a concentration of 30 nM PAPP-A2 in assays. Protein quality and representative, primary data are shown in Supplementary Fig. 4a and Supplementary Fig. 4b, respectively. Additional data for all experiments is included in Supplementary Data 2.
**f** Cleavage assays for wild type and active site PAPP-A2 mutants using IGFBP3 as a substrate. Representative, primary data is shown in Supplementary Fig. 4c. Assays shown here and elsewhere in the manuscript used 500 nM of IGFBP3. IGFBP3 was observed to be cleaved less efficiently than IGFBP5. Therefore, for assays containing IGFBP3 as a substrate shown here and in the remainder of the report, reactions were incubated for 18 h at 37 °C.
**g** Cleavage assays for wild type and anchor peptide binding deficient mutant PAPP-A2 or PAPP-A using IGFBP5 as a substrate. Protein quality is shown in Supplementary Fig. 5a and Supplementary Fig. 5b, and representative data is in Supplementary Fig. 5c. In assay results shown in Fig. 1.e–g, error bars represent the standard deviation of experiments done in triplicate.

MP domain where substrate residue S143 coordinates with the zinc[17]. Alignment of PAPP-A2 with the PAPP-A$_{BP5}$ structure suggests that the anchor peptide is within proximity of the zinc coordination site in the MP domain in PAPP-A2 (Fig. 1d, with anchor peptide in transparent orange cartoon). PAPP-A residues A449, T450, and W451 (Fig. 1d, transparent plum sticks) form a cleft around anchor peptide residue Q142, and are conserved as residues A466, T467, and W468 in PAPP-A2 (Fig. 1d, green sticks). Mutation of this patch in either enzyme abolishes activity (Fig. 1g, Supplementary Fig. 5a–c, and Supplementary Data 2), suggesting that the basic mechanism of IGFBP5 cleavage is similar for both PAPP-A2 and PAPP-A.

**The A799V patient mutation occurs in a region sensitive to mutation with a long-range connection to the active site.** The PAPP-A2 A799V (also known as A1033V in the literature when using the unprocessed protein sequence numbering) patient mutation results in loss of PAPP-A2 activity and a decrease in free IGF-1 that causes short stature[34]. PAPP-A2 A799V was previously shown to be poorly recombinantly expressed and was partially truncated compared to wildtype protein[34]. This mutant was shown to result in a loss of PAPP-A2 proteolytic activity for IGFBP3 and IGFBP5 cleavage when using supernatants from cells recombinantly expressing PAPP-A2 A799V, with similar results for both full length and truncated protein[34].

To better understand the mechanism of patient mutation inhibition, we purified PAPP-A2 A799V for in vitro cleavage assays. While we did observe lower expression of PAPP-A2 A799V compared to wildtype protein, we did not observe significant truncation of PAPP-A2 A799V as protein quality was comparable to wildtype protein (Supplementary Fig. 6a, Supplementary Fig. 3a, and Supplementary Fig. 3c). Purified PAPP-A2 A799V resulted in significant loss of IGFBP5 and IGFBP3 cleavage (Fig. 2a, Supplementary Fig. 6b, and Supplementary Data 2), in agreement with previous data from overexpression culture medium[34]. Additionally, the PAPP-A2 A799V mutant melted at a temperature of about 2 °C lower than wildtype protein by a Nano Differential Scanning Fluorimetry (NanoDSF) assay, indicating that the mutation slightly compromises thermal stability (Fig. 2b and Supplementary Data 2).

The A799V point mutation occurs in a region of the M1 domain that is over 26 Å away from the catalytic zinc as revealed by our cryo-EM structure (Fig. 2c). Analysis of our structure revealed a long-range interaction network connecting to the active site. The sidechain of A799 makes a hydrophobic

interaction with P684 and hydrogen bonds to the backbone of W375 (Fig. 2c). Residue W735 stacks with Y659, which hydrophobically interacts with P683 (Fig. 2c). Thus, P684 and P683 (Patch 1, Fig. 2c, blue script) are both directly and indirectly stabilized by A799, respectively. P683 makes a hydrophobic interaction with K472 in the MP domain with additional hydrogen bonding provided by V681 (Fig. 2c). Residue D473 hydrogen bonds to T467, the backbone amine of D471 hydrogen bonds to the backbone carbonyl oxygen of W468, and the sidechain of W468 stacks with that of W470 (Fig. 2c). Both T467 and W468 are critical for PAPP-A2 activity as we showed earlier (Fig. 1g). Thus, residues D471, K472, and D473 (Patch 2, Fig. 2c, orange script) provide stabilizing interactions that bridge the M1 and MP domains and support the patch of residues composed of A466, T467, W468, P469, and W470 (Patch 3, Fig. 2c, magenta script) that includes residues necessary for activity. We tested the importance of the stretch of bridging residues between A799 and the active site using patch mutations in P683A/P684A (Patch 1) and D471A/K472A/D473A (Patch 2). Both patch mutations resulted in significant reduction of IGFBP5 cleavage (Fig. 2d, Supplementary Fig. 6a, and Supplementary Fig. 6c, and Supplementary Data 2). Thus, the patch regions are sensitive to mutation and their long-range connection to the active site and residue A799 suggests a plausible mechanism of PAPP-A2 inactivation by the A799V patient mutation.

**Machine learning prediction structure of the PAPP-A2/ IGFBP5 complex.** Our cryo-EM structure did not reveal the C-terminal domains of PAPP-A2 or bound IGFBP5 substrate. Therefore, we pursued a machine learning complex structure prediction, using AlphaFold multimer version 2.1.2[36,37] with its default parameters to gain insight into the interaction of full-length PAPP-A2 with IGFBP5, referred to hereafter as ML-PAPP-A2/IGFBP5 (Supplementary Data 3). The model predicts that the C-terminal domains assume a *cis*-configuration with the LG domain interacting with the CCP2 domain in the same manner as the AlphaFold predicted PAPP-A2 structure [AF-Q9BXP8] (Fig. 3a, Supplementary Fig. 7a, and Supplementary Fig. 7b). Notably, the IGFBP5 anchor peptide featured a relatively low predicted local distance difference test (pLDDT) value, in agreement with the fact that no density for it was revealed by cryo-EM (Supplementary Fig. 7a). The model predicts that full length IGFBP5 is mostly composed of loops and that the anchor peptide is predicted to form a helix that binds to the MP and M1 domains of PAPP-A2 (Fig. 3a with full modeled anchor peptide

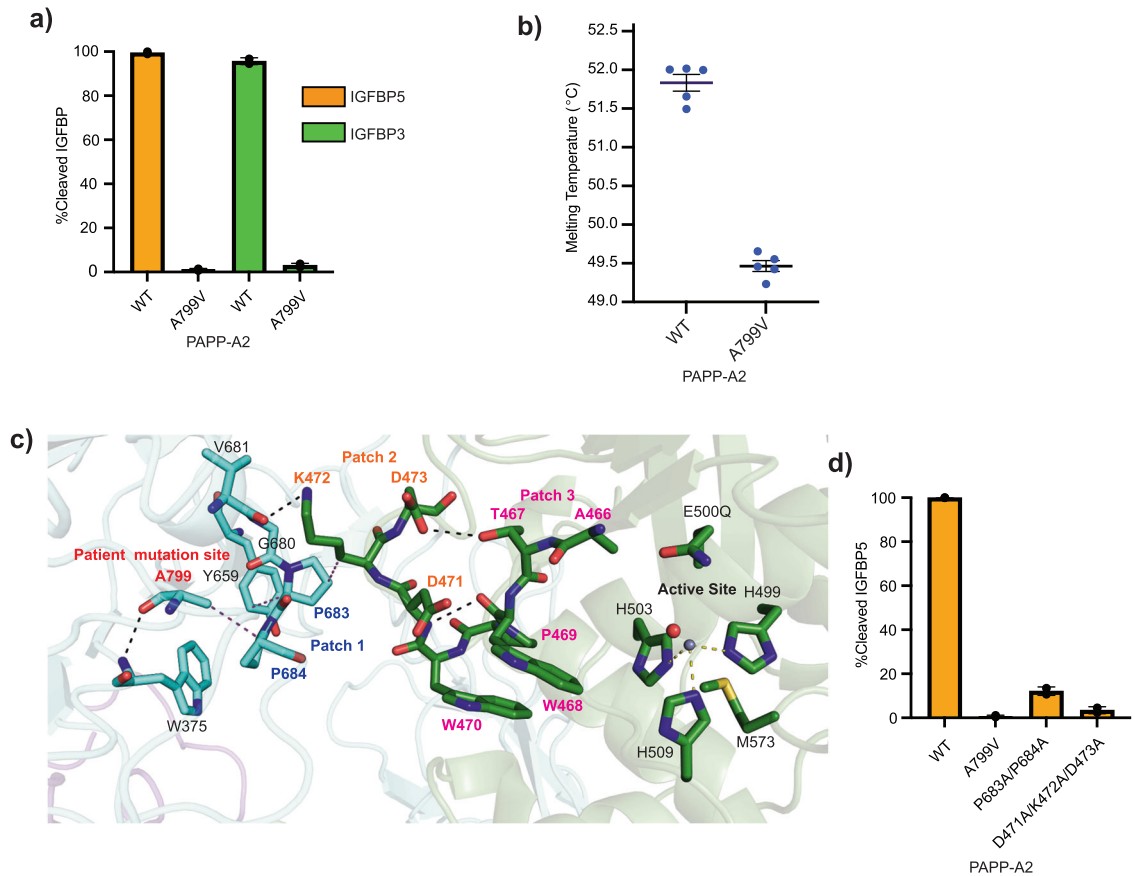

**Fig. 2 Analysis of the PAPP-A2 A799V patient mutation. a** Cleavage assay for wild type and A799V patient mutant PAPP-A2 for IGFBP5 and IGFBP3 substrates. Error bars represent the standard deviation of experiments done in triplicate. Protein quality is shown in Supplementary Fig. 6a and representative data in Supplementary Fig. 6b. **b** NanoDSF assay comparing the melting temperatures of wildtype and A799V PAPP-A2, with the average (middle line) of five experiments (individual dots) shown with error bars representing the standard deviation. **c** Zoomed-in view of the location of the patient mutation site from the PAPP-A2 cryo-EM structure and surrounding residues. Domains are color coded as in Fig. 1a. Residue A799 is shown with sticks and in red script. Patch 1 (P683, P684) in blue script, Patch 2 (D471, K472, and D473) in orange script, and Patch 3 (A466, T467, W468, P469, and W470) in pink script are shown as sticks. Hydrogen bonds, hydrophobic interactions, and metal coordination bonds are shown as black, magenta, and yellow dashed lines, respectively. **d** Cleavage assay for PAPP-A2 wild type and patch mutants for IGFBP5 substrate. Error bars represent the standard deviation of experiments done in triplicate. Protein quality is shown in Supplementary Fig. 6a and representative data in Supplementary Fig. 6c.

interactions in Supplementary Fig. 7c). In the model, anchor peptide residue K128 hydrogen bonds to S698 and this orientation is buttressed by PAPP-A2 residues W697 and P699 (Fig. 3b). In the model, IGFBP5 residue K128 makes a hydrophobic interaction with PAPP-A2 residue W675 which itself is supported by L704 and Y674 (Fig. 3b). These interactions are mirrored in the PAPP-A$_{BP5}$ cryo-EM structure (pdb 7ufg), except that Y674 is replaced by H657 in PAPP-A and there is no residue with a function corresponding to L704 of PAPP-A2 in PAPP-A (Fig. 3c). In the ML-PAPP-A2/IGFBP5 model, the side chain of E700 hydrogen bonds to anchor peptide residue K128 (Fig. 3b), but this interaction is not conserved in PAPP-A. Instead, both N683 and H781 of PAPP-A hydrogen bond to IGFBP5 anchor peptide residue K128 in the PAPP-A$_{BP5}$ cryo-EM structure[17] (Fig. 3c). Therefore, there are predicted to be both similarities and differences between the IGFBP5 anchor peptide binding pockets between PAPP-A2 and PAPP-A.

IGFBP5 is not cleaved by PAPP-A2 or PAPP-A when residue K128 is mutated (Supplementary Fig. 8a–d and Supplementary Data 2) in agreement with previous results[15,17,38], and in support of our model suggesting a similar binding location. Moreover, mutating conserved residues predicted to be involved in anchor peptide residue K128 binding (W697A/S698A/P699A or Y674A/W675A) abolishes cleavage activity of PAPP-A2 (Fig. 3d, Supplementary

Fig. 9a, b, and Supplementary Data 2). A reduction of IGFBP5 cleavage by PAPP-A is observed when the main K128 binding site (W680A/S681A/P682A) is mutated, but no effect is observed when the secondary binding site (H657A/W658A) is mutated (Fig. 3d, Supplementary Fig. 9c-d, and Supplementary Data 2), suggesting that PAPP-A is more resistant to mutation than PAPP-A2. There are other interactions in the ML-PAPP-A2/IGFBP5 model that suggest differences compared with PAPP-A. In the ML-PAPP-A2/IGFBP5 model, anchor peptide residue R137 makes a salt bridge to PAPP-A2 residue D363 in the MP domain, and anchor peptide residue R138 makes a salt bridge with M1 domain residue D687, indicating a potentially critical interaction (Fig. 3e). The importance of these residues was supported by a cleavage assay where the R136D/R137D/K138D IGFBP5 mutant could not be cleaved by PAPP-A2 (Fig. 3g, Supplementary Fig. 10a, and Supplementary Fig. 8d, Supplementary Data 2). In contrast, anchor peptide residues R137 and R138 do not contact PAPP-A directly, based on the cryo-EM structure, (Fig. 3f) and cleavage of the IGFBP5 R136/R137D/K138D mutant was significantly reduced but not abolished, indicating that these residues are less critical for binding to PAPP-A (Fig. 3g, Supplementary Fig. 10b, and Supplementary Fig. 8d, and Supplementary Data 2). Overall, although the pLDDT value of the anchor peptide is relatively low in ML-PAPP-A2/IGFBP5 model (Supplementary Fig. 7a), the structure predicts a similar binding

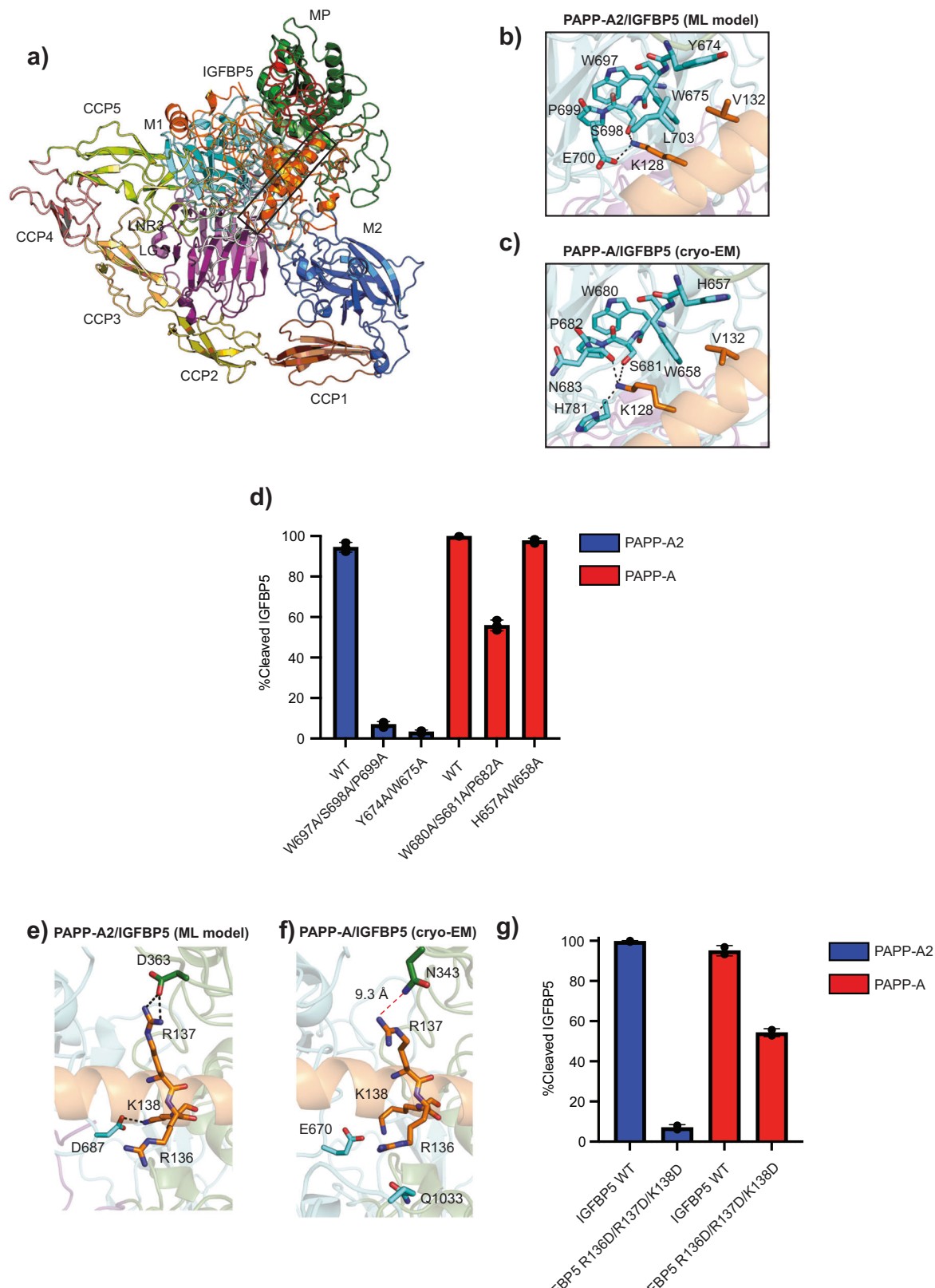

pose of IGFBP5 anchor peptide as in the PAPP-A$_{BP5}$ cryo-EM structure. Cleavage assays with mutagenesis confirmed the importance of residues predicted to form the substrate binding site from the ML-PAPP-A2/IGFBP5 model and suggested differences between IGFBP5 anchor peptide recognition mechanisms by PAPP-A and PAPP-A2.

**PAPP-A2 binds and cleaves IGFBP5 less efficiently than PAPP-A likely due to differences in the M2 domain**. PAPP-A2 and PAPP-A show a similar mode of IGFBP5 recognition yet differ in severity of cleavage reduction for most mutations tested. We hypothesize that PAPP-A2 binding to IGFBP5 is much more dynamic compared to PAPP-A. We directly compared IGFBP5

**Fig. 3 ML model and functional analysis of the IGFBP5 anchor peptide binding site in PAPP-A2. a** ML-PAPP-A2/IGFBP5 model. PAPP-A2 domains are color-coded and IGFBP5 is shown in orange. Some regions of IGFBP5 are shown as transparent cartoon to make the anchor peptide (boxed) more visible. Confidence scores are shown in Supplementary Fig. 7a. A full, zoomed-in view of the binding site is shown in Supplementary Fig. 7c. **b** Zoomed-in view of the anchor peptide binding site in the M1 domain (cyan) of the ML-PAPP-A2/IGFBP5 model. **c** Zoomed-in view of the anchor-peptide binding site in the M1 domain (cyan) from the PAPP-A$_{BP5}$ (PDB 7ufg). **d** Cleavage comparison of wildtype and mutant PAPP-A2 and PAPP-A proteins in the M1 domain anchor peptide binding site. Error bars represent the standard deviation of experiments done in triplicate. Protein quality for wildtype and mutant PAPP-A2, and a representative cleavage assay is shown in Supplementary Fig. 9a and Supplementary Fig. 9b, respectively. Protein quality for wildtype and mutant PAPP-A, and a representative cleavage assay is shown in Supplementary Fig. 9c and Supplementary Fig. 9d, respectively. **e** Zoomed-in view of the ML-PAPP-A2/IGFBP5 model around anchor peptide residues 136-138. The MP domain is in green and the M1 domain is in cyan. Black, dashed lines represent hydrogen bonds. **f** Zoomed-in view of PAPP-A$_{BP5}$ (PDB 7ufg) around anchor peptide residues 136-138. The red dashes indicate that residue R137 is too far away to interact with the MP domain. **g** Cleavage comparison of PAPP-A2 and PAPP-A on wild type and R136D/R137D/K138D IGFBP5 mutant substrate. Error bars represent the standard deviation of experiments done in triplicate. IGFBP5 substrate protein quality is in Supplemental Fig. 8d. The graphed data is representative of cleavage efficiency at a concentration of 8 nM for both PAPP-A and PAPP-A2. No additional cleavage was observed for IGFBP5 R136D/R137D/K138D at higher concentrations of PAPP-A2 (Supplementary Fig. 10a), but PAPP-A cleaved the substrate completely at concentrations above 8 nM (Supplementary Fig. 10b).

binding and cleavage by PAPP-A2 and PAPP-A to obtain a better understanding of the differences between the two enzymes. Catalytically inactive PAPP-A2 E500Q shows a 11-fold weaker binding to full-length (FL) IGFBP5 than inactive PAPP-A E483A (Fig. 4a and Supplementary Data 2) and PAPP-A2 cleaved wild type IGFBP5 38-fold less efficiently than PAPP-A (Fig. 4b, Supplementary Fig. 11, and Supplementary Data 2). We next evaluated the contribution of the C-terminal domains of PAPP-A2 to IGFBP5 cleavage. Removal of the C-terminus of the protein starting with the CCP1 domain (PAPP-A2$_{1159}$) resulted in a 2.7-fold reduction in activity (Fig. 4c, Supplementary Fig. 12a, b, and Supplementary Data 2), suggesting that this region could contribute to, but is not essential for IGFBP5 cleavage. This is not surprising as we do not observe density for these regions in the PAPP-A2 cryo-EM structure, the ML-PAPP-A2/IGFBP5 model does not predict their direct interactions with the anchor peptide, and removal of the C-terminal domains of PAPP-A also does not hinder IGFBP5 cleavage[9,17]. Interestingly, the further deletion of the M2 domain, which we did not observe in the cryo-EM structure, (PAPP-A2$_{927}$) did not result in an additional reduction in IGFBP5 cleavage (Fig. 4c, Supplementary Fig. 12a, b, and Supplementary Data 2). Key residues from the PAPP-A M2 domain that contribute to IGFBP5 anchor peptide recognition are not conserved in PAPP-A2, suggesting that the M2 domain is not necessary for substrate recognition by PAPP-A2 (Supplementary Fig. 13a–c). Dimerization interfaces are also not well conserved in PAPP-A2, underlying why PAPP-A2 is a monomer (Supplementary Fig. 1, Supplementary Fig. 14a–c). PAPP-A2 shows much lower proteolytic activity for IGFBP5 cleavage compared with PAPP-A monomer variants (PAPP-A$_{1100\ -1111*}$ and PAPP-A$_{1100\ -1135*}$) that were described previously[17], indicating that dimerization is not a key determinant for differences in IGFBP5 cleavage activity between the two enzymes (Fig. 4d, Supplementary Fig. 15a, b, and Supplementary Data 2). Instead, the dynamics of the M2 domain with regard to anchor peptide binding may be important for substrate cleavage for PAPP-A2.

Our ML-PAPP-A2/IGFBP5 model predicted a well-structured conformation of the M2 domain by interacting with M1 and the anchor peptide, however, the model just captures a single conformation and could not reflect all possible dynamics in solution. To gain deeper insight into the role of the M2 domain, we carried out three independent replicates (~2.2 μs in aggregate) of molecular dynamics (MD) simulations (Fig. 5a and Supplementary Fig. 16a) using the CHARMM36m force field[39] on the ML-PAPP-A2/IGFBP5 model. Our analysis of the MD simulations revealed that PAPP-A2 tends to adopt predominant 'open' conformations (Fig. 5a). Remarkably, our MD simulations showed that the M2 domain is highly flexible and undergoes a

conformational change from a closed state, with a 54 Å distance between M1-M2 domains, predicted by AlphaFold multimer (Fig. 3a), to largely open states, with a 62.9 ± 5.3 Å distance between M1-M2 domains (Fig. 5b, M1-M2, and Supplementary Fig. 16a), indicating its dynamic nature. This distinct spatial separation between M2 and M1 subdomains exerts a substantial impact on the interactions between the IGFBP5 anchor peptide and M2 domain. We find that the M2 domain undergoes a displacement of ~14 Å away from the IGFBP5 anchor peptide, expanding from 38.0 Å in the closed state to 51.7 ± 7.1 Å in open states (Fig. 5b, anchor peptide-M2, and Supplementary Fig. 16a), leading to limited contribution of the M2 domain to the binding of the anchor peptide (Fig. 5a and Fig. 5c, Anchor peptide-M2 distance). The prevalence of predominant 'open' conformations of PAPP-A2 (Fig. 5c) suggests that the cleavage of IGFBP5 may not proceed as efficiently as it would with a tightly closed dimeric PAPP-A structure, and this hypothesis is supported by our binding and activity data (Fig. 4a and Fig. 4b) and may explain why removing the M2 domain did not directly affect IGFBP5 cleavage (Fig. 4c).

To eliminate the potential impact of our selected force field on the observed behavior of the M2 domain, we decided to perform two independent MD simulations (~2.4 μs in aggregate) using alternative force fields: a99SB-disp[40] and AMBER-14[41] (Supplementary Fig. 16b–g). These MD simulations also find that the M2 domain exhibits a propensity to separate from the M1 domain, notably at distances of 67.8 + /-2.4 Å (Supplementary Fig. 16b) for a99SB-disp and 57.0 Å +/-0.1 Å (Supplementary Fig. 16c) for AMBER-14. Nonetheless, in contrast to the other two selected force fields, AMBER-14 results in a modest degree of separation between the M2 and M1 domains (Supplementary Fig. 16c and Supplementary Fig. 16e), with an opening of 2.7 Å compared to the AlphaFold multimer structure (Fig. 3a). We attribute this behavior to concerns previously raised regarding the excessive stabilization of protein-protein interactions by AMBER-14[42]. However, the divergence of the M2 domain from the M1 domain by a margin of ~3 Å holds significant implications, effectively interrupting possible interactions between the M2 domain and the anchor peptide (Supplementary Fig. 16e), expanding the distance between the pair by 10.5 Å compared to ML-PAPP-A2/IGFBP5 model, measured at 48.5 + /-0.1 Å for AMBER-14 (Supplementary Fig. 16c). Thus, regardless of the choice of force field, our extensive MD simulations consistently suggest that the PAPP-A2 M2 domain is dynamic.

Together, our data show that although IGFBP5 is a substrate for both PAPP-A and PAPP-A2, its cleavage efficiency by the two metalloproteases is different. Our computational modeling suggests that the M2 domain in PAPP-A2 could exist as

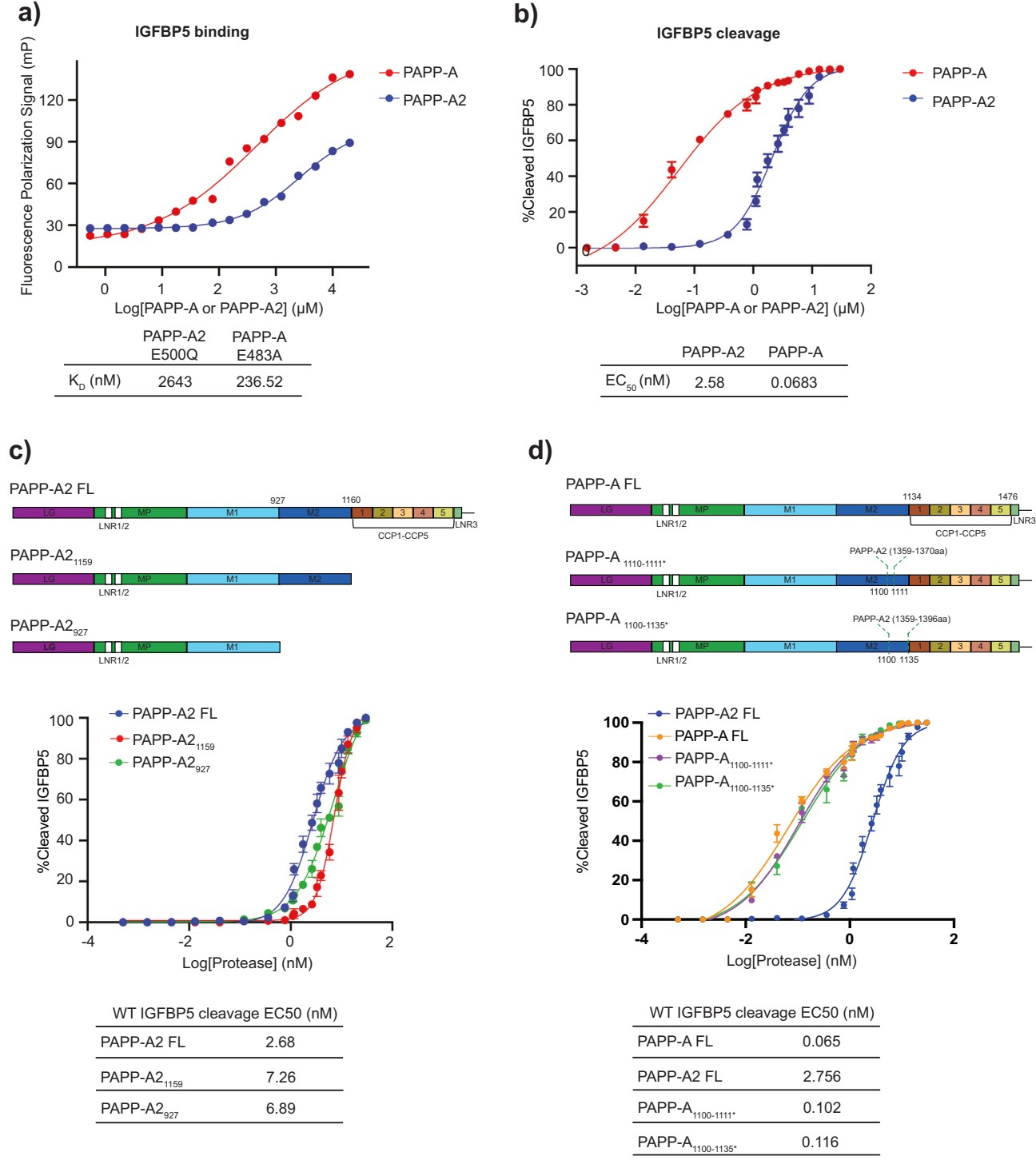

**Fig. 4 PAPP-A2 and PAPP-A feature differences in the mechanism of IGFBP5 cleavage. a** Comparison of FL IGFBP5 binding to the catalytically inactive version of PAPP-A2 (E500Q) and PAPP-A (E483A) using a fluorescent polarization assay. Fluorescently labeled FL IGFBP5 was used as the substrate and catalytically inactive PAPP-A2 and PAPP-A proteins were used to prevent substrate cleavage. **b** Comparison of PAPP-A2 and PAPP-A IGFBP5 cleavage using equimolar enzyme concentrations. Representative data is shown in Supplementary Fig. 11. **c** Schematic diagram of PAPP-A2 truncation proteins and cleavage assay comparison with wild type PAPP-A2. Protein quality is shown in Supplementary Fig. 12a and representative data in Supplementary Fig. 12b. **d** Schematic diagram of PAPP-A monomer variants with M2 domain PAPP-A2 insertions and cleavage assay comparison with FL PAPP-A2 and PAPP-A. Protein quality is shown in Supplementary Fig. 15a and representative data in Supplementary Fig. 15b. For all experiments in this figure error bars represent the standard deviation of experiments done in triplicate.

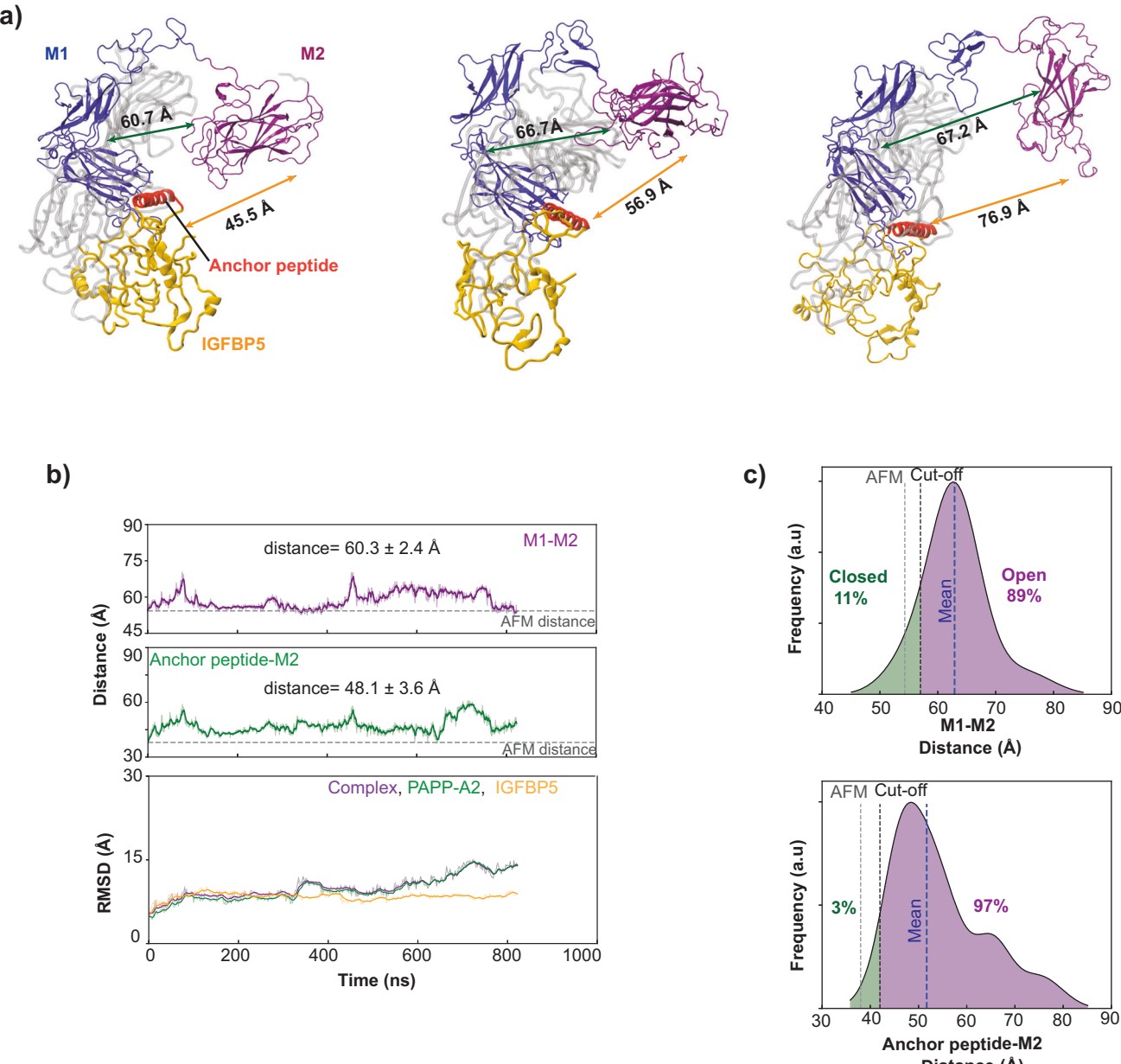

**Fig. 5 MD-simulations feature a highly dynamic M2 domain in PAPP-A2. a** Open conformations of PAPP-A2 structures emerged from three independent replicates of MD simulations using the CHARMM36m force field. The snapshots were captured at time points: 600 ns of replicate 1 (left), 600 ns of replicate 2 (middle), and 628 ns of replicate 3 (right). Conformational changes in PAPP-A2, crucial for substrate recognition, were evaluated based on the distances between the M1 domain (center of mass of C$\alpha$ atoms of residues: 610-927) and the M2 domain (center of mass of C$\alpha$ atoms of residues: 953-1160), as well as the M2 domain and the anchor peptide (center of mass of C$\alpha$ atoms of residues: 120-143). For the sake of clarity, the remainder of the C-terminus of PAPP-A2 (residues 1167 to the end) is not depicted. **b** Graphs depicting simulation results: M2-M1 distance (top), M2-anchor peptide distance (middle), and RMSD values for C$\alpha$ atoms (bottom). RMSD values were measured relative to the AlphaFold multimer (AFM) construct at the start of the simulations ($t = 0$ ns). Exponential moving averages over the trajectory are shown as solid lines. For each replicate, average and standard deviation were computed using block averaging, using the final 400 ns frames to reduce the effects of initial transients. Eight blocks of 50 ns frames were used for measurement. **c** Distribution histograms: distance between M2 and M1 domains (top), and distance between the M2 domain and the anchor peptide (bottom). Histograms are based on the last 400 ns frames of all production simulations. To estimate distance distributions, a Gaussian kernel density estimator was applied using Python's scikit-learn package, using a covariance factor of 3 Å. The overall average and standard deviation for three replicates are as follows: M1-M2 distance = 62.9 ± 5.3 Å and M2-anchor peptide distance = 51.7 ± 7.1 Å. The AlphaFold multimer (AFM) distances are shown as dashed lines in the plots. The cut-off distances were calculated as AFM distance + 0.5 * standard deviation, to capture a region that extends beyond the AFM close state while considering the spread of the distribution. Simulation run graphs for replicate 2 and 3 are shown in Supplementary Fig. 16a.

predominantly 'open' conformations which are unfavored for IGFBP5 recognition. The calculated dynamics of the M2 domain support PAPP-A2's monomeric nature and its lower potency for IGFBP5 cleavage.

## Discussion

Circulating IGFs are primarily associated with either IGFBP3 or IGFBP5 and insulin-like growth factor-binding protein complex acid labile subunit (ALS), and bound IGFs are therefore prevented from activating growth factor signaling[43–45]. Sequestered IGFs are liberated by cleavage of IGFBP proteins by PAPP-A2 and PAPP-A. While recent publications have described the structure of apo PAPP-A, PAPP-A bound to the anchor peptide region of IGFBP5, and PAPP-A bound to STC2 and proMBP proteins[17–19], much less is known about PAPP-A2. Unraveling the non-redundant role of PAPP-A2 is important to understanding its modulation for IGF signaling. Therefore, we combined several techniques to more holistically characterize PAPP-A2. We showed with cryo-EM that the overall N-terminal LG, MP, and M1 domains of PAPP-A2 are similar to those of PAPP-A but that PAPP-A2 is a monomer (Fig. 1b and Fig. 1c) and validated this with biochemical analysis (Supplementary Fig. 3a–c). Our cryo-EM structure of PAPP-A2 did not reveal density for the LNR1 and LNR2 regions, the M2 domain, or the CCP1-CCP5 domains likely due to flexibility and possibly due to lack of PAPP-A2 dimerization (Fig. 1b).

Importantly, the PAPP-A2 structure revealed the location of the A799V patient mutation in relation to the active site (Fig. 2c). We showed that PAPP-A2 A799V is slightly less thermally stable than wildtype protein (Fig. 2b) but not unfolded. We also identified patches of residues leading to the zinc binding site and connected to residue A799 that resulted in protein inactivation when mutated (Fig. 2d), suggesting a possible long-range destabilization caused by the A799V patient mutation.

We used machine learning to predict interactions between PAPP-A2 and IGFBP5 (Fig. 3a, Fig. 3b, and Fig. 3e) and combined with our cryo-EM structure, we used mutagenesis with cleavage assays to support these predictions by showing that PAPP-A2 is more vulnerable to mutation than PAPP-A (Fig. 3d and Fig. 3g). Our biochemical assays also revealed that PAPP-A2 has weaker IGFBP5 binding and cleavage efficiency compared to PAPP-A (Fig. 4a and Fig. 4b). We used MD simulations based on our ML model and the calculated result suggested that the M2 domain of PAPP-A2 predominantly adopts an 'open' conformation and may play a different role in substrate binding compared to that of PAPP-A (Fig. 5a). This agrees with our cryo-EM structure in which the M2 domain density is not observed, possibly due to it being dynamic. The M2 domain in PAPP-A plays a critical role in substrate recognition and dimerization[17], but these regions are not conserved in PAPP-A2 (Supplementary Fig. 13a and Supplementary Fig. 14a) and we showed that deletion of the M2 domain results in only a modest reduction in PAPP-A2 activity (Fig. 4c). These studies suggest that the M2 domain in PAPP-A2 has a less important role in substrate recognition compared to PAPP-A. It is worth noting that the MD simulation was performed using the ML-PAPP-A2-IGFBP5 model as the template, and we validated hypotheses derived from the model with functional data. We note that there are limitations of this analysis from not having a starting experimental structure and that future experimental structural studies would be helpful to fully validate our model.

We also note that the mechanism of cleavage of IGFBP3 by PAPP-A2 is still not clear. Sequence alignment (Supplementary Fig. 17a) and structural prediction (Supplementary Fig. 17b) suggests that IGFBP3 contains a similar anchor peptide region as

IGFBP5 so it is plausible that it is recognized by PAPP-A2 in a generally similar manner. However, recognition must be unique to PAPP-A2 as IGFBP3 cannot be cleaved by PAPP-A. In vivo, IGFBP3 is bound to IGF-1 and ALS, and a cryo-EM structure of this complex was recently reported but did not reveal the anchor peptide region of IGFBP3[46]. PAPP-A2 has a much slower efficiency for IGFBP3 cleavage compared with IGFBP5[46]. Yet PAPP-A2 is critical for IGFBP3 cleavage because IGFBP3 levels are lower in patients with the A799V mutation which results in reduced growth[34]. Therefore, other factors may be required for PAPP-A2 mediated cleavage of IGFBP3 in vivo. IGFBP3 has been shown to be cleaved by both ADAM12 and thrombin in vitro[46,47], possibly suggesting alternative modes of processing.

IGF signaling is a tightly regulated pathway. Other than IGF sequestration by IGFBP proteins, the pathway is also regulated by STC1 and STC2 proteins that inhibit PAPP-A and PAPP-A2, and proMBP that inhibits PAPP-A. A recent cryo-EM structure visualized that both PAPP-A monomers cooperate to bind to two proMBP monomers[19] in agreement with prior data obtained from biochemical experiments[48,49]. PAPP-A2 does not bind proMBP in serum[6,14] so it is unlikely that proMBP plays a role in PAPP-A2 inhibition. The structural basis of PAPP-A inhibition by STC2 was also recently reported by two groups and indicated that a dimer of STC2 is bound by cooperation between LNR1, LNR2, and LNR3 regions and M1 domains of each monomer of PAPP-A[18,19]. While PAPP-A2 is inhibited by STC2[12], PAPP-A2 is notably a monomer when unbound (Fig. 1b, Supplementary Fig. 2a, b, Supplementary Fig. 3a–c), so it is unclear if it binds monomeric or dimeric STC2, and its mode of inhibition by STC1 is also unknown. Therefore, the precise mechanism of PAPP-A2 regulation by other proteins is an area of future research.

In summary we used a variety of tools to probe the mechanism of substrate recognition by PAPP-A2. We note that the true nature of transiently stable complexes such as PAPP-A2 and IGFBP5 is unlikely to be determined by experimental approaches alone, and that a combination of machine learning and MD-simulation helped us to obtain a more complete mechanistic characterization of this system. Overall, the findings of our study provide critical insight into PAPP-A2, an enzyme that is less studied than its paralog PAPP-A, yet very relevant for human growth and possibly other areas of human health. Future work will need to be conducted to further differentiate the roles of PAPP-A2 and PAPP-A in IGFBP5 cleavage in vivo.

## Methods

**DNA construct design and mutagenesis**. The generation of mammalian-based expression constructs for wildtype PAPP-A, PAPP-A2, IGFBP5, and PAPP-A$_{1100\ -1111*}$ and PAPP-A$_{1100\ -1135*}$ was described previously[17]. PAPP-A2 and PAPP-A constructs contain a C-terminal FLAG tag for use in purification, and IGFBP5 constructs contain an N-terminal 6xHis tag. PAPP-A2, PAPP-A, and IGFBP5 plasmids containing mutants, deletions, and insertions generated for use in this work were introduced by standard PCR-based site-directed mutagenesis by Quintara Biosciences (Hayward, CA). All mutation sites were confirmed by sequencing.

**Protein expression and purification**. PAPP-A2, PAPP-A, and IGFBP5 were expressed in Expi293F (Thermo Fisher, cat. #A14527) cells and secreted into the culture medium. Cells were pelleted by centrifugation and the supernatant was harvested. PAPP-A and PAPP-A2 were purified from the supernatant by Anti-FLAG chromatography (GenScript Anti-DYKDDDDK G1 Affinity resin, Cat. #L00432) in 1X PBS (154 mM NaCl, 1.06 mM KH$_2$PO$_4$, Na$_2$HPO$_4$, Corning product #21-040-CM) and eluted

with 1 mg/mL 3X-DYKDDDDK peptide (APEXBIO, product #A6001) diluted in 1X PBS. PAPP-A and PAPP-A2 proteins were concentrated and injected into a Superose 6 Increase 10/300 column (Cytiva, product #29091596) run in 1XPBS on an AKTA Pure 25 M (cytiva) system running Unicorn software 7 (Cytiva). PAPP-A2 eluted as a monomer while PAPP-A eluted as a dimer. Peak fractions were pooled, concentrated, snap frozen in liquid nitrogen, and stored at -80 °C until use. PAPP-A2 mutants and truncations, and PAPP-A mutants and PAPP-A/PAPP-A2 hybrids were purified in an identical manner as wild type proteins. Wildtype IGFBP5 and IGFBP5 mutants were purified with Ni-NTA agarose resin (Qiagen, product #30210) and eluted in 0.5 M imidazole pH 8.0, 0.5 M NaCl in 1X PBS buffer. The eluted fractions were concentrated and run on a Superdex 200 Increase 10/300 GL column (Cytiva, product # 28990944). Peak fractions were pooled, concentrated, and stored at −80 °C until further use.

Recombinant human IGFBP3 (Abcam, Cat #ab280941) was reconstituted in 1X PBS (Corning product #21-040-CM) to a concentration of 1 mg/ml then aliquoted and stored at −80 °C for future use.

**Cleavage assays**. In vitro cleavage reactions were carried out in a total reaction volume of 30 µL in 1X PBS (Corning product #21-040-CM). IGFBP5 or IGFBP3 was used as the substrate at a final concentration of 500 nM for proteolytic cleavage reactions. For all assays, equimolar concentrations of PAPP-A2 and PAPP-A were compared. Molar concentrations were based on monomeric PAPP-A2 or dimeric PAPP-A. Proteolytic reactions were performed at 37 °C for 4 h. All reactions were quenched by the addition of 0.1 mM EGTA (Fisher Scientific, product #AAJ60767AD). The quenched reactions were applied to Bolt 4–12% Bis-Tris SDS-PAGE gels (Invitrogen, product #NW04122Box) run under reducing conditions in 1X MES buffer (diluted from 20X stock, Invitrogen, product #B000202) and stained with Instant Blue Coomassie Protein Stain (abcam product #ab119211). Cleavage efficiency was determined by integrating substrate (IGFBP3 or IGFBP5) band intensities with Image Lab (Bio-Rad,Version 6.1.0) and calculating the percentage of cleavage against intact substrate controls. Experiments were completed in triplicate. Average and standard deviations of replicates were calculated with Microsoft Excel (Version16.59).

For all assays, serial titration of protease was performed to determine the cleavage efficiency, and this data is shown Supplementary Data 2. Graphs showing single titration points in the main text used 30 nM of enzyme, unless indicated otherwise, where cleavage efficiency by PAPP-A2 and PAPP-A was ~100%. For graphs with multiple titration points, we performed serial dilutions of PAPP-A2, PAPP-A, or variants to determine accurate $EC_{50}$ values, for comparison of cleavage efficiency. The percentage of cleaved substrate was plotted against the log concentration of protease to determine $EC_{50}$ values by fitting the percent cleavage vs PAPP-A concentration to a non-linear regression dose response model using Prism (GraphPad Software, Prism version 9.1.2).

**Fluorescence polarization binding assays**. Recombinant IGFBP5 was labeled with FAM-maleimide, 6-isomer (Lumiprobe, Cat#24180) following the manufacturer's recommended protocol. IGFBP5 was reconstituted to a concentration of 3 mg/mL using 1X PBS, pH 7.4 (Corning product #21-040-CM). Tris-carboxyethylphosphine (TCEP) dissolved in molecular biology grade water at a stock concentration of 1 mM was added to the IGFBP5 solution to a final concentration of 0.1 mM. The sample was kept at room temperature for 20 min to reduce disulfide bonds. FAM-maleimide, 6-isomer dissolved in DMSO at

1 mg/mL was added to the sample and allowed to incubate at 4 °C overnight. Excess dye and reducing agent were then removed by gel filtration using a Superdex 200 Increase 10/300 GL column (Cytiva, 28-9909-44) equilibrated in 1X PBS, pH 7.4.

PAPP-A2 binding to FAM-labeled IGFBP5 was measured by fluorescence polarization on a CLARIOstar plate reader (BMGLabtech, Cat#0430-101) using 384-well fluorescence assay plates (Corning, ProductNumber: 4514). Measurements were made using an optical path consisting of a 482-16 nm excitation filter, LP 504 dichroic mirror, and a 530-40 nm emission filter. FAM-labeled IGFBP5 at a final concentration of 5 nM in 1X PBS was used for gain, focal height, and baseline adjustments. For direct binding measurements, PAPP-A2 was serially diluted from 10 µM to 0.5 nM onto the assay plate in 1X PBS. The time point was initiated with the addition of FAM-labeled IGFBP5 to a final concentration of 5 nM per well. Polarization values were taken every 65 s on the plate reader at 220 flashes per read.

**Non-reducing PAGE and native PAGE protein analysis**. Wild type PAPP-A2 and A799V mutant were analyzed by SDS-PAGE and blue native-PAGE. Bolt Bis-Tris Plus Mini Protein Gels, 4–12%, 1.0 mm (Invitrogen) were used for SDS-PAGE analysis. For SDS-PAGE under reducing conditions, proteins were mixed with NuPAGE™ LDS Sample Buffer (4X) and 2-Mercaptoethanol (500 mM) and boiled for 5 min. at 95 °C. For SDS-PAGE under non-reducing conditions, proteins were mixed with 1X NuPAGE LDS Sample Buffer (Thermo Fisher, catalogue #NP0007) and run directly on the gel without any boiling. SDS-PAGE gels were run for 28 min at 200 V using NuPAGE 1X MES SDS Running Buffer (Invitrogen, product #B000202). The gels were stained using Instant Coomassie Blue Stain (Abcam, catalogue #ab119211) for 2–3 h and destained overnight using MilliQ Water.

For Native PAGE, NuPAGE 3–8% Tris-Acetate Mini Protein Gels (Invitrogen) were used for analysis. Blue native-PAGE gels were run for 2–2.5 h at 125 V using 10x Novex Tris-Glycine Native Running Buffer (Life Technologies) according to the manufacturer's instructions. Samples were prepared in 1X Tris-Glycine Native Sample Buffer and 5% Coomassie. 1X Novex Tris-Glycine Native Running Buffer with 1% Coomassie was used in the cathode until the gel was run one-third of the way and replaced with only 1X Running buffer for the rest of the run. The gels were stained using Colloidal Blue Staining Kit (Invitrogen, catalogue #LC6025) and destained using MilliQ water. The SDS-PAGE and Native gels were imaged using BioRad imager ChemiDocMP.

**NanoDSF analysis**. Thermal shift assays were performed on wildtype PAPP-A2 and PAPP-A2 A799V using Prometheus Panta NT.48 from NanoTemper technologies by measuring the intrinsic dual-UV fluorescence change in tryptophan and tyrosine residues in proteins at emission wavelengths of 330 and 350 nm. The ratio of the recorded emission intensities (Em350nm/Em330nm), which represents the change in tryptophan fluorescence intensity as well as the shift of the emission maximum to higher wavelengths (red-shift) or lower wavelengths (blue-shift) was plotted as a function of the temperature. The fluorescence intensity ratio and its first derivative were calculated and determined to be the melting temperature, with the manufacturer's software (PR.Panta Control and PR.Panta Analysis). The samples were loaded using capillaries in a volume of 20 µL on Prometheus Panta NT.48 from NanoTemper Technologies. Concentrations of 10 µM of wildtype PAPP-A2 and 8 µM of PAPP-A2 A799V were subjected to thermal change from 25 °C–95 °C with a ramp rate of 0.5 °C/min.

**Cryo-EM PAPP-A2 grid preparation and data acquisition**. A total 76.78 µL of purified PAPP-A2 E500Q protein at a concentration of 1.2 mg/mL was incubated with 23.3 µL of IGFBP5 at a concentration of 0.7 mg/mL at a 1:1 molar ratio on ice for 40 min., centrifuged at 13500xg at 4 °C for 5 min., and then injected into a Superdex 200 column and run on a micro AKTA system (Cytiva). The peak fraction was used for cryo-EM grid preparation. An aliquot of 4 µL of sample was applied onto a glow-discharged 400 mesh grid (Quantifoil Au R1.2/1.3) that was supported with a thin layer of graphene oxide, blotted with filter paper for 3.5 s, and plunge-frozen in liquid ethane using a Thermo Fisher Vitrobot Mark IV. Cryo-EM micrographs were collected on a 300 kV Thermo Fisher Krios G3i electron microscope equipped with a K3 direct detection camera and a Bio-Quantum image filter (GIF: a slit width of 20 eV). The micrographs were collected at a calibrated magnification of x105,000, yielding a pixel size of 0.669 Å at a super resolution counting mode. In total, 4734 micrographs were collected at an accumulated electron dose of 50 e-Å-2 s-1 on each micrograph that was fractionated into a stack of 32 frames with a defocus range of -1.0 µm to −2.0 µm.

**EM data processing**. Beam-induced motion correction was performed on the stack of frames using MotionCorr2[50]. The contrast transfer function (CTF) parameters were determined by CTFFIND4[51]. A total 4,734 good micrographs were selected for further data processing using cryoSPARC[52]. Particles were auto-picked by the auto-picking program in cryoSPARC, followed by 3 rounds of reference-free 2D classifications. Next, 269,763 particles were selected from good 2D classes and were subjected to a round of 3D classification using a reconstruction of PAPP-A2 as a starting model. Four converged 3D classes with a feature containing PAPP-A2 were selected for a final round of 2D classification, in which 220,575 particles were selected for a final round of 3D refinement. A tighter mask created by RELION[53] using relion_mask_create was applied when performing 3D refinement. Before mask creation, we manually erased most of the peripheral noise of the 3D map that was generated from cryoSPARC. The –ini_threshold was set to 0.28, while –extend_inimask and –width_soft_edge" were both set to 4. The mask enclosed the entirety of the map, and no additional density was present that could account for a second copy of PAPP-A2. The use of the mask yielded a final reconstruction at a global resolution of 3.13 Å based on the Gold-Standard Fourier Shell Correlation (GSFSC) criterion of 0.143[54]. The local resolution was then calculated on the final density map.

**Model building and refinement**. The signal sequence and pro-peptide regions (residues 1-234) from the AlphaFold2[36,37] PAPP-A2 model (AF-Q9BXP8-F1) were removed as they are not present in the purified protein. The truncated model was then fit to the map using Phenix Dock. The structure was refined using Phenix RealSpaceRefinement[55,56] alternating with manual building in Coot[57]. No density was observed for the M2, CCP1-5 or the LNR 1, 2 and 3 domains so these were truncated from the structure. Analysis and model validation for the structure were performed using Coot and the Phenix validation tool.

**Machine learning-based PAPP-A2/IGFBP5 model**. AlphaFold multimer version 2.1.2[36,37] with its default parameters was used to generate the ML-PAPP-A2/IGFBP5 model. The mature versions of PAPP-A2 with signal peptide and pro-peptide regions removed and IGFBP5 with signal peptide region removed were as inputs for AlphaFold multimer.

**Molecular dynamics simulations**. We used the ML predicted PAPP-A2/ IGFBP5 complex as the starting point. We then inserted a zinc in the catalytic site of PAPP-A2 and restrained its distance to three residues (H499, H503, and H509) during the whole calculations. Hence, we placed three harmonic restraints at 3.0 Å between (i): Zn-H499 (NE2), (ii): Zn-H503(NE2), and (iii) Zn-H509(NE2) with a force constant of 0.48 kcal mol-1 Å-2. Subsequently, we immersed the complex in a box (118.0 Å * 142.6 Å * 126.0 Å) of water (61 K molecules) and neutralized the system with excess NaCl to achieve a physiological salt concentration of 150 mM. To refine the construct, the system was minimized using 500 steps of energy minimization according to the steepest descents algorithm incorporated in GROMACS[58]. The optimization was followed by an MD simulation in a canonical ensemble, where the system was heated gradually from 0 K to 310 K in 200 ps. Then, an MD simulation in an isobaric-isothermal ensemble was carried out for 1 ns with maintaining the pressure at 1 bar to relax the simulation box. During these whole pre-equilibration steps, the positional restraints were placed on all heavy atoms and Zn ions using 47.8 kcal.mol-1Å2, which were progressively reduced to 0 kcal.mol-1Å2 for the final equilibration step. Subsequently, three separate replicates of MD simulations (~2.2 µs in aggregate) in an isobaric-isothermal were performed to equilibrate the PAPP-A2/IGFBP5 construct.

The Charmm36m parameter set[39] was used to describe PAPP-A2, IGFBP5, and ions, while water was described using the CHARMM TIP3P model. The temperature was maintained at 310 K using a velocity-rescale[59] thermostat with a damping constant of 1.0 ps for temperature coupling and the pressure was controlled at 1 bar using a Parrinello-Rahman barostat algorithm[60] with a 5.0 ps damping constant for the pressure coupling. Isotropic pressure coupling was used during this calculation. The Lennard-Jones cutoff radius was 12 Å, where the interaction was smoothly shifted to 0 after 10 Å. Periodic boundary conditions were applied to all three directions. The Particle Mesh Ewald algorithm[61] with a real cutoff radius of 10 Å and a grid spacing of 1.2 Å was used to calculate the long-range coulombic interactions. A compressibility of 4.5 × 10-5 bar-1 was used to relax the box volume. In all the above simulations, water OH bonds were constrained by the SETTLE algorithm[62]. The remaining H-bonds were constrained using the P-LINCS algorithm[63]. All MD simulations were carried out using GROMACS-2021[58,63], with constrained MD simulations aided by PULMED-2.8.0[64]. We carried out two independent simulations (~2.4 µs in aggregate) using a99SB-disp[40] force field with the recommended TIP4P-D water model[65] and also with AMBER-14[41] with the recommended TIP3P water model[66] to fairly evaluate the outcome of computational modeling. The full analysis and results produced by these simulations were shown in Supplemental Fig. S16. We used Visual Molecular Dynamics (VMD-1.9.4)[67] to visualize and analyze the simulation trajectories.

**Structural visualization**. Molecular graphics were prepared with PyMOL software (The PyMOL Molecular Graphics System, version 2.5.2 Schrödinger, LLC.) or UCSF ChimeraX (version 1.4)[68,69]. UCSF ChimeraX is developed by the Resource for Biocomputing, Visualization, and Informatics at the University of California, San Francisco, with support from National Institutes of Health R01-GM129325 and the Office of Cyber Infrastructure and Computational Biology, National Institute of Allergy and Infectious Diseases.

**Sequence alignment**. Protein sequence alignment was completed with Clustal Omega[70,71].

**Statistics and reproducibility**. Data are presented as mean values ± SD (standard deviation) calculated using Microsoft Excel 2022 (version 16.59) and GraphPad Prism 9 (version 9.1.2). Derived statistics correspond to analysis of averaged values across independent replicates. For the percent cleavage activity curves, non-linear regression dose-response model was used to determine the EC50 values.

**Reporting summary**. Further information on research design is available in the Nature Portfolio Reporting Summary linked to this article.

## Data availability

The coordinates for PAPP-A2 have been deposited in the Protein Data Bank under the accession number 8sl1. The cryo-EM map for PAPP-A2 has been deposited into the Electron Microscopy Data Bank with the accession number EMD-40571. The Protein Data Bank validation report is included as Supplementary Data 1. Experimental source data is included as Supplementary Data 2. The ML-PAPP-A2/IGFBP5 PDB file is available as Supplementary Data 3. For MD-simulation, the full data set of PDB files was deposited on GCP bucket and can be accessed through the link below: https://console.cloud.google.com/storage/browser/pappa2_igfbp5_md_simulations_public?hl=en&project=calico-public-data&pageState=(%22StorageObjectListTable%22:(%22f%22:%22%255B%255D%22))&prefix=&forceOnObjectsSortingFiltering=false.

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

## Acknowledgements

We thank Xiaodan Ni of Shuimu BioSciences, Ltd. for cryo-EM data collection and processing. We thank Leanne Chan of Calico for helpful suggestions for experiments. We thank Aaron Nile and Jiyeon Lee from Calico for reviewing our manuscript and providing helpful suggestions. We thank Yuliya Kutskova and Jiefei Geng from AbbVie for providing suggestions for our work.

## Author contributions

J.S. expressed and purified proteins and conducted all functional assays. A.M. performed MD-based simulations and refined models. R.A.J. fit and refined the PAPP-A2 cryo-EM structure with supervision by V.S.S. J.X. performed machine learning modeling for PAPP-A2/IGFBP5. K.A.K. and J.C.K.W. expressed proteins and assisted with purification. M.B. and Q.H. jointly conceptualized and supervised the project with additional supervision by D.E. G.K. and R.J.S. provided critical input. M.B., J.S., and A.M. generated figures. M.B., A.M., J.S., R.A.J., and Q.H. wrote the manuscript with suggestions from all authors.

## Competing interests

J.S, A.M, J.X., J.C.K.W., G.K., R.J.S., D.E., M.B., and Q.H. are employees of Calico Life Sciences, LLC. K.A.K. is a contractor for Calico Life Sciences. LLC. R.A.J. and V.S.S. are employees of AbbVie and may hold AbbVie stock.
