## [Peer Review File · Communications Chemistry]

Reviewers' comments:

Reviewer #1 (Remarks to the Author):

This manuscript describes the first structure of PAPP-A2 and highlights the similarities and differences with the existing PAPP-A:IGFBP-5 structures that were published recently by this group and others. Using a combination of cryo-electron microscopy, molecular dynamics and AlphaFold the researchers propose a plausible model for the mechanism of IGFBP-5 cleavage by PAPP-A2 (noting that my expertise does not lie in cryo-EM, MD or AlphaFold).

This manuscript is very well written and contributes to our understanding of the mechanism of action of this family of enzymes, and in particular the mechanism of IGFBP-5 cleavage. The structure reveals a probable explanation for the lower substrate cleavage efficiency compared to PAPP-A. An explanation of the impact of a clinically relevant PAPP-A2 A799V mutation on the enzyme's function is provided and an interesting long-range stabilization network is described. Unfortunately, the determinants of substrate specificity remain unclear (eg IGFBP-5 versus IGFBP-3). Nonetheless, this body of work represents a significant contribution to the understanding of how IGF action is regulated through enzyme cleavage of IGFBPs.

Comments/suggested corrections:

Figure 1d. The zinc and water spheres are on top of H509 in this orientation and are difficult to see – the orientation in Fig S2e is better. Consider using the same orientations in both Fig 1d and Fig. S2e. It would also be good to mention the IGFBP-5 concentration in Fig. 1e legend and state this is the concentration used throughout.

Figure 3 legend: Should “Prediction scores” really be “Confidence scores” as in Fig. S6a?

Figure 3a: It may be easier for the reader if the same orientation is presented in Fig. 3a and Fig. 1b and c.

Figure S2: There should be a legend for Fig. S2g

Figure S3b and c: provide units on figures for molecular weight standards. As molecular weight markers are not provided for each gel panel it is not clear whether the E500Q mutant is running smaller on the gel or whether this is just an alignment issue. Preferably each panel would have its own markers.

Figure S4: please state which IGFBP is being used in Fig. S4C

Figure S8 legend “(related to Fig. S3e)” should be “(related to Fig. 3e)”. Please state which IGFBP is being used in Fig. S8C

Figure S10 please provide molecular weight markers.

Figure S13 Please check the labelling of the residues in Fig. S13b and c. For example, F1101' in Fig. S13b

but L1101 (red) in Fig. S13c. There may be some mix up between the PAPP-A and PAPP-A2 numbering.

Reviewer #2 (Remarks to the Author):

Sridar et al. present a structural, biochemical, and modeling study related to PAPP-A2, a metalloprotease known to cleave IGF binding proteins (IGFBP3 and IGFBP5) for modulating IGF signaling. Specifically, authors employ cryo-electro microscopy to determine the structure at 3.13 Å resolution. Based on this structure, authors also carry out molecular dynamics simulations and modeling using structural prediction servers such as AlphaFold2, especially to rationalize effect of a specific mutation (A799V/A1033V depending on sequence numbering convention). Overall, the manuscript data is described in detail. Some comments for authors to consider are listed below:

1. In my opinion, a major weakness of the manuscript is overemphasis and stretch of conclusions drawn from computationally predicted models, especially known limitations of MD simulations including the shorter time-scales authors have probed for a protein of this large size, without any convergence metrics from simulations, without any testing of the variation in the protein force-fields. While it would be considerable work for authors to undertake a thorough MD based analysis, at the very least, it should be made clear throughout the manuscript that ideas emerging from limited modeling and simulations are speculative. There are too many places where the cleavage mechanism or IGFBP5 interaction or conformational change is mentioned as overly accurate or explaining missing links. At one place authors talk about a chain like or domino effect and suggest it as a “straightforward” explanation of lack of activity, only to step back by mentioning the key unknown aspect whether the mutation even leads to properly folded protein or not. For this and similar claims made, mostly in the latter half of the manuscript, authors should carefully review sentences and wording to more precisely describe, what can be reasonably articulated in terms of claims/conclusions drawn from modeled and simulated data. The discussion section should summarize at a single place all major conclusions drawn from simulations and their limitations and the need for future experimental studies to test speculative ideas.
2. Authors claim a conformational change of 1.4 Å as significant, especially when the mutant model is not experimentally determined. It is common in predicted models to obtain RMSDs of ~4-5 Å especially with the uncertainty in positioning of the residue sidechains. Ion placements are even more erroneous. Again, please describe the rationale for calling it significant or remove such wording.
3. In the para just before the discussion section, “confirmations” should be “conformations.” Please check for similar issues at other places.
4. Authors should provide coordinates of all modeled data and sample MD trajectory data (e.g., 100-500 conformers of protein per MD run, after excluding solvent). The input files for conducting these MD simulations should also be provided via data sharing platforms such as github or zenodo and a link to those data should be added within the manuscript.

Reviewer #3 (Remarks to the Author):

Below examples serve to illustrate a general lack of precision with regard to aim/research questions,

reasoning, and citation that must be corrected before this can be published. There are also several examples of omission of earlier published results as well as technical shortcomings.

Abstract

-please be precise about the research questions (here an later in the ms). The patient mutation mentioned in the abstract is already known from the literature to cause inactivity of PAPP-A2. The mechanism of this is a hypothesis resulting from the current study (based on theoretical analyses). Correct wording might be, e.g., "A patient mutation, previously shown to cause PAPP-A2 inactivity, may cause destabilization...".

(Note that the authors say in the discussion: "Therefore, the chain-like effect provides a straightforward explanation for the lack of activity in the PAPP-A2 patient mutation, while we note that we cannot completely rule out other effects of the A799V mutation on protein folding.)

Note: Also, as mentioned below, Dauber 2016 showed that the A799V variant in fact 1) expresses poorly in a recombinant system, and 2) undergoes (partial) autocleavage. These earlier finding must be mentioned.

-own previous report of PAPP-A structure is irrelevant (in particular because there are now three published PAPP-A structures in the literature). Be precise about the research question.

-please be precise about what parts of the structure is reported (not just N-terminal part). Also not e.g. LNR1.

Introduction

-number of residues in mature protein is incorrectly specified as 1524??

-sentence "Both PAPP-A2 and PAPP-A contain an N-terminal Laminin-G (LG) domain and a catalytic metalloprotease (MP) domain that itself contains two Lin12-Notch repeats (LNR1 and LNR2) (Boldt et al., 2006)": Why is reference made only to Boldt 2006?? Boldt 2004 should be placed here also. Place references accurately, which appears to be journal style. MANY similar inaccuracies throughout the ms.

-sentence: "Unlike PAPP-A that forms a trans-homodimer, PAPP-A2 exists as a monomer (Judge et al., 2022; Overgaard et al., 2001)." Missing citation = Weyer 2007. Weyer 2007 is the first demonstration that PAPP-A dimerizes in trans. Later both Judge 2022 and Kobberø 2022 confirmed this. Again, place references accurately to allow the reader to go back in the literature.

-that PAPP-A2 exists as a monomer is NOT experimentally supported. Discovery paper (Overgaard 2001) shows that PAPP-A2 subunits DO NOT form a COVALENT SS-based dimer. However, Weyer 2007 shows that IN NATIVE PAGE, PAPP-A and PAPP-A2 migrates similarly, thus PAPP-A2 migrates as a dimer. SEC of the current study does not provide any info about size in this regard. Of importance, this has implications throughout the manuscript, including reasoning MANY places. Serious revision is required.

-also, must be discussed in relation to how refinement mask was define. Did the refinement mask cover a sub-volume only? Not clear at all.

-relating to the two previous points: 2D classification figure of Figure 2S suggests symmetry??

-sentence “Also, PAPP-A binds to cell surface glycosaminoglycans (GAGs) via its CCP4 and CCP5 domains”: Is NOT correct. PAPP-A binds GAG via CCP3 and CCP4 (Laurson 2002b), not CCP4 and CCP5 as stated. Based on the cited paper it cannot be concluded that “PAPP-A2 mainly exists in circulation”. For a discussion of that, see e.g. PMID 36718521.

-sentence: Our previous report together with other studies suggested various key determinants of PAPP-A required for cleavage of IGFBP4 but not for IGFBP5: 1) LNR center formation, 2) an interaction between the LG and CCP2 domains, and 3) trans-dimerization (Boldt et al., 2004; Judge et al., 2022; Weyer et al., 2007). It is very misleading to use this kind of wording. Only the LG-CCP2 point refers to “Our previous report” (= Judge 2022), and it is unclear what is meant by that – to what extent this is experimentally supported. Reference for point 1 is Boldt 2004. Reference for point 3 is Weyer 2007.

-sentence: “Interestingly, the A799V point mutation site is far from the predicted proteolytic site of PAPP-A2, and its inactivation mechanism is unclear.”: It should be stated specifically here what was found in the Dauber paper: That, A799V has no catalytic activity towards IGFBP3 and 5. That the recombinant protein expresses much less efficiently compared to wild-type PAPP-A. AND that it is susceptible to autocleavage (that might be inactivating). Completely left out??

Results

-active site inactivated PAPP-A2 has previously been generated and analyzed (discovery paper = Overgaard 2001). It is incorrect to say “in agreement with analogous results with PAPP-A”. Reference to actual PAPP-A2 paper is required.

-Figure 1e+f: Active site-inactivated PAPP-A2 previously shown (Overgaard 2001). Here a main figure?

-Figure 3d: K128D shown previously. Here a main figure?

-sentence: “Therefore, the chain-like effect provides a straightforward explanation for the lack of activity in the PAPP-A2 patient mutation, while we note that we cannot completely rule out other effects of the A799V mutation on protein folding.” : As mentioned above, compromised recombinant expression as reported in Dauber 2016 should be referred to also here. Further to mention: Dauber reported intramolecular cleavage of the mutant, which may also cause disturbed structure/activity. Therefore, it is critical to show the A799V variant under BOTH reducing and nonreducing conditions.

-sentence p7: “..and removal of the C-terminal domains of PAPP-A also does not hinder IGFBP5 cleavage (Judge et al., 2022).” This is a key finding about PAPP-A. Made in 2004 (Boldt 2004). Elaborated on in

several following papers, most recently in Kobberø 2022. Citing Judge is narrow-minded and with no respect towards the scientific literature.

-indicate clearly for all PAPP-A2 SDS-PAGE gels whether the gels were run under reducing or nonreducing conditions.

-a lot of excessive speculation in Results, which does not belong there. Also, in general it is often difficult to follow exactly what the authors are suggesting as experimental support.

-may not be clear to the reader that some results are predictions based on predictions...

Discussion:

-Sentence: "Sequestered IGFs are liberated by cleavage of IGFBP proteins by PAPP-A2, PAPP-A, and other proteases": Which other proteinases? In vitro, in vivo? Evidence? See e.g. PMID 36718521.

p9, "A recent structure revealed that both PAPP-A monomers cooperate to bind to two proMBP monomers (Zhong et al., 2022)." Wrong citation. Not revealed by a recent study. Stoichiometry/and binding was determined/analyzed in Overgaard 2003 (PMID 12421832) and Glerup 2005 (PMID 15647258).

-sentence top of p9 "...the substrate recognition residues": Not at all defined to a level that justifies reference like that. Potential over-interpretation of effects of mutation.

Methods:

Cleavage assay: What happened after the gels were run? Staining? Blotting? Detection? Details lacking. PBS not defined here (or elsewhere? Calcium concentration???)

p17, relating to points about monomer/dimer: "PAPP-A/PAPP-A2 Proteins were concentrated and injected into a Superose 6 Increase 10/300 column (Cytiva) run in 1XPBS. PAPP-A2 eluted as a monomer while PAPP-A eluted as a dimer.": This CANNOT be concluded. There are no figures showing co-runs, and the column was not even calibrated. Statement is based on assumption only. Again, native PAGE shows differently as mentioned above.

Please see our response to the reviewers' comments with references to line numbers in the text and highlighted in yellow in the manuscript.

Reviewers' comments:

Reviewer #1 (Remarks to the Author):

This manuscript describes the first structure of PAPP-A2 and highlights the similarities and differences with the existing PAPP-A:IGFBP-5 structures that were published recently by this group and others. Using a combination of cryo-electron microscopy, molecular dynamics and AlphaFold the researchers propose a plausible model for the mechanism of IGFBP-5 cleavage by PAPP-A2 (noting that my expertise does not lie in cryo-EM, MD or AlphaFold).

This manuscript is very well written and contributes to our understanding of the mechanism of action of this family of enzymes, and in particular the mechanism of IGFBP-5 cleavage. The structure reveals a probable explanation for the lower substrate cleavage efficiency compared to PAPP-A. An explanation of the impact of a clinically relevant PAPP-A2 A799V mutation on the enzyme's function is provided and an interesting long-range stabilization network is described.

Unfortunately, the determinants of substrate specificity remain unclear (eg IGFBP-5 versus IGFBP-3). Nonetheless, this body of work represents a significant contribution to the understanding of how IGF action is regulated through enzyme cleavage of IGFBPs.

We are glad that this reviewer is pleased with our study and thank this reviewer for these encouraging remarks. Please see our responses below. Note that some figures have been renumbered in the revision.

Comments/suggested corrections:

Figure 1d. The zinc and water spheres are on top of H509 in this orientation and are difficult to see – the orientation in Fig S2e is better. Consider using the same orientations in both Fig 1d and Fig. S2e.

We agree and Fig. 1d has been revised to depict the view in the original Fig. S2e, which has been renumbered as Fig. S2h in the revision.

It would also be good to mention the IGFBP-5 concentration in Fig. 1e legend and state this is the concentration used throughout.

In the legend for Fig. 1e, we now state that 500 nM of IGFBP5 was used here and throughout the report (lines 333-334).

Figure 3 legend: Should “Prediction scores” really be “Confidence scores” as in Fig. S6a?

This has been corrected to “confidence scores” in the legend of Fig. 3a (lines 360-361) as was described in the original Fig. S6a that has been renumbered as Fig. S7a.

Figure 3a: It may be easier for the reader if the same orientation is presented in Fig. 3a and Fig. 1b and c.

We agree with the reviewer. Fig. 1b and Fig. 1c are now shown in a similar orientation with the MP domain pointed upward. Fig. 3a is now shown in the same orientation. We have also revised Fig. 3b and Fig. 3c to show the same orientation, with the anchor peptide represented diagonally.

Figure S2: There should be a legend for Fig. S2g

We thank the reviewer for catching this mistake. This has been corrected and now a legend is included for all parts of Fig. S2 (lines 442-452).

Figure S3b and c: provide units on figures for molecular weight standards. As molecular weight markers are not provided for each gel panel it is not clear whether the E500Q mutant is running smaller on the gel or whether this is just an alignment issue. Preferably each panel would have its own markers.

We thank the reviewer for this suggestion. Molecular weight markers have been included on all gels in these figures, which were renumbered to Fig. S4b and Fig. S4c.

Figure S4: please state which IGFBP is being used in Fig. S4C

Wild-type IGFBP5 is used, and this is now stated in the legend to this figure (lines 475-476). Please note that this figure has now been renumbered to Fig. S5c.

Figure S8 legend “(related to Fig. S3e)” should be “(related to Fig. 3e)”. Please state which IGFBP is being used in Fig. S8C

We have made these corrections, but first we need to explain that these figures have undergone renumbering. Fig. S8 has been renumbered to Fig. S9 and Fig. 3e has been renumbered to Fig. 3d. Thus, the title for Fig. S9 is now stated to be related to Fig. 3d (lines 506-507). Fig. S8c has been renumbered as Fig. S9d and wild-type IGFBP5 is stated to have been used in the legend to this figure (line 512).

Figure S10 please provide molecular weight markers.

This figure has been renumbered to Fig. S11 and molecular weight markers have been added. Note, we now include molecular weight markers on all SDS-PAGE gels that are shown.

Figure S13 Please check the labelling of the residues in Fig. S13b and c. For example, F1101' in Fig. S13b but L1101 (red) in Fig. S13c. There may be some mix up between the PAPP-A and PAPP-A2 numbering.

We thank the reviewer for pointing out this error and it has been corrected. Please note that Fig. S13 has been renumbered to Fig. S14.

Reviewer #2 (Remarks to the Author):

Sridar et al. present a structural, biochemical, and modeling study related to PAPP-A2, a metalloprotease known to cleave IGF binding proteins (IGFBP3 and IGFBP3) for modulating IGF signaling. Specifically, authors employ cryo-electro microscopy to determine the structure at 3.13 Å resolution. Based on this structure, authors also carry out molecular dynamics simulations and modeling using structural prediction servers such as AlphaFold2, especially to rationalize effect of a specific mutation (A799V/A1033V depending on sequence numbering convention). Overall, the manuscript data is described in detail.

We thank the reviewer for these comments. Please see our responses below. Note that some figures have been renumbered in the revision.

Some comments for authors to consider are listed below:

1. In my opinion, a major weakness of the manuscript is overemphasis and stretch of conclusions drawn from computationally predicted models, especially known limitations of MD simulations including the shorter time-scales

authors have probed for a protein of this large size, without any convergence metrics from simulations, without any testing of the variation in the protein force-fields. While it would be considerable work for authors to undertake a thorough MD based analysis, at the very least, it should be made clear throughout the manuscript that ideas emerging from limited modeling and simulations are speculative.

We thank the reviewer for these helpful comments. We now include a more thorough treatment of our MD results in the revision (lines 224-252) and directly address the reviewer's concerns below.

To comprehensively address the concerns arising from the computational modeling outcomes, we have undertaken a series of rigorous steps for revision. We first extended the time scale of the three replicates of MD simulations using CHARMM36m force field, accumulating a total of 2.2 microsecond. Subsequently, we conducted a comprehensive analysis of the metrics pertinent to conformational changes in PAPP-A2—specifically, the M1-M2 distance, M2-anchor peptide distance, and RMSD—all of which play a pivotal role in substrate recognition.

The RMSD variation analysis indicates that ~300 ns was sufficient for each of the three MD simulation replicates to reach equilibrium. To minimize the impact of initial transients, we exclusively employed the final 400 ns frames from each replicate for calculating averages, standard deviations, and generating comprehensive histogram plots.

As illustrated in Figure 5b-c and Figure S16a (also shown below), the outcomes of these three simulations consistently depict a prevalence of open conformations in PAPP-A2. Notably, the M2 domain exhibits a substantial departure from the M1 domain, resulting in an 8.6 Å displacement. This pronounced opening effectively disrupts all direct interactions between the M2 domain and the anchor peptide, leading to an outward displacement of the M2 domain by an additional 11.7 Å. Thus, these results suggest a limited contribution of the M2 domain to anchor peptide binding.

To eliminate the potential impact of our selected force field on the observed behavior of the M2 domain, we extended our study by performing two additional independent MD simulations (~2.4 μ s in aggregate) using alternative force fields: a99SB-disp and AMBER-14. These MD simulations also find that the M2 domain exhibits a propensity to separate from the M1 domain, notably at distances of 67.8 \pm 2.4 Å for a99SB-disp and 57.0 Å \pm 0.1 Å for AMBER-14 as shown in Fig. S16b. Notably, in contrast to the other two selected force fields, the choice of AMBER-14 force field results in the modest degree of separation between the M2 and M1 domains (Figure S16b-c), with an opening of 2.7 Å compared to the AlphaFold multimer structure (Figure 3a). We attribute this behavior to concerns previously raised (Abriata et al. 2021, PMID: 34025949) regarding the excessive stabilization of protein-protein interactions by AMBER-14. However, the divergence of the M2 domain from the M1 domain by a margin of ~3 Å holds significant implications, effectively

interrupting any direct interactions between the M2 domain and the anchor peptide, expanding the distance between the pair by 10.5 Å [measured at 48.5 +/-0.1 Å for AMBER-14, shown in Figure S16b (also shown below). Thus, regardless of the choice of force field, our extensive MD simulations consistently suggest a reduced role of the M2 domain in the cleavage of the anchor peptide.

These results are in agreement with our cleavage assay data, showing that the deletion of M2 domain did not affect PAPP-A2 activity for IGFBP5 cleavage (Fig. 4c)

(a)

(b)

(c)

Figure 5. MD-simulations feature the high dynamics of M2 domain in PAPP-A2

a) Open conformations of PAPP-A2 structures emerged from three independent replicates of MD simulations using the CHARMM36m force field. The snapshots were captured at time points: 600 ns of replicate 1 (*left*), 600 ns of replicate 2 (*middle*), and 628 ns of replicate 3 (*right*).

Conformational changes in PAPP-A2, crucial for substrate recognition, were evaluated based on the distances between the M1 domain (center of mass of C α atoms of residues: 610-927) and the M2 domain (center of mass of C α atoms of residues: 953-1160), as well as the M2 domain and the anchor peptide (center of mass of C α atoms of residues: 120-143). For the sake of clarity, the remaining region of the PAPP-A2 C-domain (residues 1167 to the end) is not depicted. **b)** Graphs depicting simulation results: M2-M1 distance (*top*), M2-anchor peptide distance (*middle*), and RMSD values for C α atoms (*bottom*). RMSD values were measured relative to the AlphaFold multimer (AFM) construct at the start of the simulations (t=0 ns). Exponential moving averages over the trajectory are shown as solid lines. For each replicate, average and standard deviation were computed using block averaging, using the final 400 ns frames to reduce the effects of initial transients. Eight blocks of 50 ns frames were used for measurement. **c)** Distribution histograms: distance between M2 and M1 domains (*top*), and distance between the M2 domain and the anchor peptide (*bottom*). Histograms are based on the last 400 ns frames of all production simulations. To estimate distance distributions, a Gaussian kernel density estimator was applied using Python's scikit-learn package, using a covariance factor of 3 Å. The overall average and standard deviation for three replicates are as follows: M1-M2 distance = 62.9 ± 5.3 Å and M2-anchor peptide distance = 51.7 ± 7.1 Å. The AlphaFold multimer (AFM) distances are shown as dashed lines in the plots. The cut-off distances were calculated as AFM distance + 0.5 * standard deviation, to capture a region that extends beyond the AFM close state while considering the spread of the distribution. Simulation run graphs for replicate 2 and 3 are shown in Fig. S16a.

(a)

(b)

(c)

(d)

a99SB-disp

(e)

AMBER-14

(f)

a99SB-disp

(g)

AMBER-14

Figure S16. Characterizing the structural dynamics of PAPP-A2 in complex with IGFBP5 protein with MD simulations using different force fields (related to Fig. 5a)

a) CHARMM36M force field with the CHARMM TIP3P water model. **b)** a99SB-disp force field with TIP4P-D water model. **c)** AMBER-14 with TIP3P water model. For each force field, graphs depict MD simulation results for two replicates. Conformational changes in PAPP-A2, crucial for substrate recognition, were evaluated based on the distances between the M1 domain (center of mass of C α atoms of residues: 610-927) and the M2 domain (center of mass of C α atoms of residues: 953-1160), as well as the M2 domain and the anchor peptide (center of mass of C α atoms of residues: 120-143). RMSD values were also measured relative to the AlphaFold multimer (AFM) construct at the start of the simulations (t=0 ns). Exponential moving averages over the trajectory are shown as solid lines. For each replicate, average and standard deviation were determined through block averaging, using the final 400 ns frames of MD simulations with the CHARMM36M force field, 1200 ns frames of MD simulation with the a99SB-disp force field, and 900 ns frames of MD simulations with AMBER-14 to mitigate initial transients. We used blocks of 50 ns frames for measurement. **d)** Open conformation of PAPP-A2 obtained using a99SB-disp force field at t=1400 ns. **e)** Open conformation of PAPP-A2 obtained using AMBER-14 force field at t=1000 ns. **f)** Distribution histogram for a99SB-disp force field. **g)** Distribution histogram for AMBER-14 force fields. For f) and g) the distance between M2 and M1 domains is shown in the top panel and the distance between the M2 domain and the anchor peptide is shown in the bottom panel. Histograms are based on the last 1200 ns, and 900 ns frames of production simulations with a99SB-disp and AMBER-14 force fields, respectively. To estimate distance distributions, a Gaussian kernel density estimator was applied using Python's scikit-learn package, using a covariance factor of 3 Å for a99SB-disp force field and 1 Å for AMBER-14. The AlphaFold multimer (AFM) distance is shown as dashed lines in the plots. The cut-off distances were selected exactly from those calculated for the simulations with the CHARMM36M force field for fair comparison, where the cut-off was determined as the AFM distance + 0.5 * standard deviation.

There are too many places where the cleavage mechanism or IGFBP5 interaction or conformational change is mentioned as overly accurate or explaining missing links.

The reviewer's comment is well noted. We assume that the reviewer is referring to our analysis of the ML-based PAPP-A2 model ("IGFBP5 interaction") and the MD simulations for PAPP-A2 ("conformational change") in this comment. We regret if we caused any confusion regarding how results were derived. In the revision, we take greater care to differentiate experimental results (the PAPP-A2 cryo-EM structure and mutagenesis studies) from ML models of MD simulations.

Results from computation are now explicitly stated to be *predictions* from models, and key examples of our rewording are illustrated below:

Lines 166-168:

“The model predicts that the C-terminal domains assume a cis-configuration with the LG domain interacting with the CCP2 domain in the same manner as the AlphaFold predicted PAPP-A2 structure [AF-Q9BXP8] (Fig. 3a, Fig. S7a, and Fig. S7b).”

Lines 170-174:

“The model predicts that full length IGFBP5 is mostly composed of loops and that the anchor peptide is predicted to form a helix that binds to the MP and M1 domains of PAPP-A2 (Fig. 3a with full modeled anchor peptide interactions in Fig. S7c). In the model, anchor peptide residue K128 hydrogen bonds to S698 and this orientation is buttressed by PAPP-A2 residues W697 and P699 (Fig. 3b). In the model, IGFBP5 residue K128 makes a hydrophobic interaction with PAPP-A2 residue W675 which itself is supported by L704 and Y674 (Fig. 3b).”

Lines 179-180:

“Therefore, there are predicted to be both similarities and differences between the IGFBP5 anchor peptide binding pockets between PAPP-A2 and PAPP-A.”

Please note, that we also validated the likelihood of several of these interactions (please see Fig. 3d and Fig. 3g) and we have revised our wording to take care to not represent the modeling results as being overly accurate, and key examples of wording changes are illustrated below:

Lines 183-185:

“Moreover, mutating conserved residues predicted to be involved in anchor peptide residue K128 binding (W697A/S698A/P699A or Y674A/W675A) abolishes cleavage activity of PAPP-A2 (Fig. 3d, Fig. S9a, and Fig. S9b).”

Lines 188-192:

“There are other interactions in ML-PAPP-A2/IGFBP5 model that suggest differences compared with PAPP-A. In the ML-PAPP-A2/IGFBP5 model, anchor peptide residue R137 makes a salt bridge to PAPP-A2 residue D363 in the MP domain, and anchor peptide residue R138 makes a salt bridge with M1 domain residue D687, indicating a potentially critical interaction (Fig. 3e). The importance of these residues was supported by a cleavage assay where the R136D/R137D/K138D IGFBP5 mutant could not be cleaved by PAPP-A2 (Fig. 3g, Fig. S10a, and Fig. S8d).”

Lines 195-199:

“Overall, although the pLDDT value of the anchor peptide is relatively low in ML-PAPP-A2/IGFBP5 model (Fig. S7a), the structure predicts a similar binding pose of IGFBP5 anchor peptide as in the PAPP-A_{BP5} cryo-EM structure. Cleavage assays with mutagenesis confirmed the importance of residues predicted to form the substrate binding site from the ML-PAPP-A2/IGFBP5 model and suggested differences between IGFBP5 anchor peptide recognition mechanisms by PAPP-A and PAPP-A2.”

We feel that these revisions accurately represent the results from the ML model without presenting the data as being overly accurate. Please also note that we provide a more thorough treatment of MD simulations (“conformational change”) is discussed in the previous comment and rewording regarding these results is discussed in a later comment.

At one place authors talk about a chain like or domino effect and suggest it as a “straightforward” explanation of lack of activity, only to step back by mentioning the key unknown aspect whether the mutation even leads to properly folded protein or not.

We thank the reviewer for this very helpful comment. We agree that our explanation may have been an oversimplification and have made significant revisions to this section of the manuscript. We have removed mention of the “straightforward explanation” wording to avoid confusion. We included nano DSF data that shows that the PAPP-A2 A799V mutant is slightly less stable than WT protein yet not completely unfolded. To avoid oversimplification, we have also removed mention of a “domino effect” and instead state in lines 157-159:

“Thus, the patch regions are sensitive to mutation and their long-range connection to the active site and residue A799 suggests a plausible mechanism of PAPP-A2 inactivation by the A799V patient mutation”

We have also changed our title to “Cryo-EM structure of PAPP-A2 and mechanism of substrate recognition” (line 1) to avoid overemphasis on the patient mutation. We feel that these revisions allow us to accurately represent our data and to avoid overinterpretation.

For this and similar claims made, mostly in the latter half of the manuscript, authors should carefully review sentences and wording to more precisely describe, what can be reasonably articulated in terms of claims/conclusions drawn from modeled and simulated data.

We thank the reviewer for the suggestion. We have significantly optimized the computational modeling by increasing the timescale and tested additional force fields to address the concerns raised earlier (please see reply to comment #1). We also reworded several regions of the manuscript to avoid putting too much emphasis on results from modeling or MD simulations. We now explicitly state when analysis is derived from a *model* whenever the PAPP-A2/IGFBP5 ML model is discussed (as illustrated previously). When discussing MD data, we make the distinction that these are calculated results, as illustrated below:

Lines 255-256:

“The calculated dynamics of the M2 domain support PAPP-A2’s monomeric nature and its lower potency for IGFBP5 cleavage.”

Lines 281-283:

“We used MD simulations based on our ML model and the calculated result suggested that the M2 domain of PAPP-A2 predominantly adopts an ‘open’ conformation and may play a different role in substrate binding compared to that of PAPP-A (Fig. 5a).”

The discussion section should summarize at a single place all major conclusions drawn from simulations and their limitations and the need for future experimental studies to test speculative ideas.

We thank the reviewer for this comment and agree with this suggestion. Please see lines 281-290 of our discussion regarding MD results, how they relate to our experimental results and ML model, and limitations and future work below:

“We used MD simulations based on our ML model and the calculated result suggested that the M2 domain of PAPP-A2 predominantly adopts an ‘open’ conformation and may play a different role in substrate binding compared to that of PAPP-A (Fig. 5a). This agrees with our cryo-EM structure in which the M2 domain density is not observed, possibly due to it being dynamic. The M2 domain in PAPP-A plays a critical role in substrate recognition and dimerization (Judge et al. 2022), but these regions are not conserved in PAPP-A2 (Fig. S13a and Fig. S14a) and we showed that deletion of the M2 domain results in only a modest reduction in PAPP-A2 activity (Fig. 4c). These studies suggest that the M2 domain in PAPP-A2 has a less important role in substrate recognition compared to PAPP-A. It is worth noting that the MD simulation

was performed using the ML-PAPP-A2-IGFBP5 model as template, and we validated hypotheses derived from the model with functional data. We note that there are limitations of this mechanism from not having a starting experimental model and that future experimental structural studies would be helpful to fully validate our model.”

2. Authors claim a conformational change of 1.4 Å as significant, especially when the mutant model is not experimentally determined. It is common in predicted models to obtain RMSDs of ~4-5 Å especially with the uncertainty in positioning of the residue sidechains. Ion placements are even more erroneous. Again, please describe the rationale for calling it significant or remove such wording.

We thank the reviewer for this comment and apologize for our oversight. We have decided to remove the modeling results for the patient mutation and instead just present the experimental data as we do not want to misrepresent the result as being overly accurate.

The results that we initially presented in our computational modeling specifically focused on elucidating the impact of the patient mutation and are derived from a de novo structural characterization conducted using MOE software. The rationale behind this approach lies in our aim to capture the initial stages of structural change, whether it involves a subtle destabilization within the catalytic site or an alteration in the folding behavior of PAPP-A2 due to the A799V mutation. Our strategy revolves around investigating the immediate vicinity of A799V, recognizing it as the potential trigger point for these structural alterations.

We understand the raised question about the relatively low RMSD value of 1.4 Å. It's important to note that this value pertains to a specific set of residues (12 residues), encompassing a localized region of the PAPP-A2 protein. In certain contexts, even minor RMSD changes of this nature can hold significant functional implications. A relevant illustration is observed in class A GPCRs, where a slight shift of ~1 Å within the NPxxY motif (Dror et al. 2011, PMID: 34025949) can distinguish between active and inactive conformations.

3. In the para just before the discussion section, “confirmations” should be “conformations.” Please check for similar issues at other places.

We thank the reviewer for pointing out this mistake. We have fixed this issue and checked to make sure that “conformations” is used throughout the manuscript.

4. Authors should provide coordinates of all modeled data and sample MD trajectory data (e.g., 100-500 conformers of protein per MD run, after excluding solvent). The input files for conducting these MD simulations should also be provided via data sharing platforms such as github or zenodo and a link to those data should be added within the manuscript.

We will gladly deposit the structure files and simulation input files (force field parameters and simulation parameters) into a public depository at the end of this review process so that others can access our data to reanalyze the results.

Reviewer #3 (Remarks to the Author):

Below examples serve to illustrate a general lack of precision with regard to aim/research questions, reasoning, and citation that must be corrected before this can be published. There are also several examples of omission of earlier published results as well as technical shortcomings.

We thank the reviewer, and these suggestions are very well noted. We apologize for unintentionally omitting citations to prior research and we have corrected these omissions. We have made significant revisions to add clarity to avoid confusions of our aims. We have addressed all technical shortcomings brought up by this reviewer. Please see the detailed responses that follow. Please also note that some figures have been renumbered in the revision.

Abstract

-please be precise about the research questions (here an later in the ms). The patient mutation mentioned in the abstract is already known from the literature to cause inactivity of PAPP-A2. The mechanism of this is a hypothesis resulting from the current study (based on theoretical analyses). Correct wording might be, e.g., “A patient mutation, previously shown to cause PAPP-A2 inactivity, may cause destabilization...”.

We did not intend to imply that we reported a new patient mutation. We had cited the reference for the patient mutation in our introduction section (lines 72-74). We agree with the reviewer that we could have made this clear in the abstract. We

corrected our wording in the abstract as follows: “Our structure together with functional studies provides insight into a previously reported patient mutation that inactivates PAPP-A2 in a distal region of the protein” (lines 19-20).

(Note that the authors say in the discussion: “Therefore, the chain-like effect provides a straightforward explanation for the lack of activity in the PAPP-A2 patient mutation, while we note that we cannot completely rule out other effects of the A799V mutation on protein folding.).

We thank the reviewer for this helpful comment and agree that we should not oversimplify the effect of the patient mutation. We have removed wording indicating a “straightforward explanation,” “domino effect,” or “chain-like effect” to avoid oversimplification. We now include protein nano thermal shift data that shows the PAPP-A2 A799V mutant is slightly less thermal stable than WT but not completely unfolded (Fig. 2b).

We now used the following wording to describe the effect of the patient mutation and our analysis of the patch mutations in lines 144-159 and summarized here:

“The A799V point mutation occurs in a region of the M1 domain that is over 26 Å away from the catalytic zinc as revealed by our cryo-EM structure (Fig. 2c). Analysis of our structure revealed a long-range interaction network connecting to the active site... Both patch mutations resulted in significant reduction of IGFBP5 cleavage (Fig. 2d, S6a, and S6c). Thus, the patch regions are sensitive to mutation and their long-range connection to the active site and residue A799 suggests a plausible mechanism of PAPP-A2 inactivation by the A799V patient mutation.”

Note: Also, as mentioned below, Dauber 2016 showed that the A799V variant in fact 1) expresses poorly in a recombinant system, and 2) undergoes (partial) autocleavage. These earlier finding must be mentioned.

Based on reviewer's suggestion, we have revised this section to include citation to Dauber 2016 as follows (lines 133-136):

“PAPP-A2 A799V was previously shown to be poorly recombinantly expressed and was partially truncated compared to wildtype protein (Dauber et al. 2016). This mutant was shown to result in a loss of PAPP-A2 proteolytic activity for IGFBP3 and IGFBP5 cleavage when using supernatants from cells recombinantly expressing PAPP-A2 A799V, with similar results for both full length and truncated protein (Dauber et al. 2016).”

-own previous report of PAPP-A structure is irrelevant (in particular because there are now three published PAPP-A structures in the literature). Be precise about the research question.

We thank the reviewer for this comment and are sorry if our wording caused confusion. We did not mean to imply that our previous structure of PAPP-A/IGFBP5 anchor peptide was the only structure that has been reported. We have now changed our abstract to read (lines 15-17):

“We previously reported the homodimeric structure of PAPP-A in complex with the anchor peptide region of its substrate IGFBP5 and structures of PAPP-A in complex with inhibitory proteins have been reported as well.”

We also cite all published PAPP-A cryo-EM structures in our introduction section as follows (lines 48-50):

“PAPP-A was first experimentally shown to form a *trans*-homodimer (Weyer et al. 2007) and was later visualized as a dimer by cryo-EM reconstruction in the apo form and when bound to IGFBP5 (Judge et al. 2022), and when bound to inhibitory proteins STC2 and proMBP (Kobbero et al. 2022; Zhong et al. 2022).”

In the discussion section of our original manuscript, we cited the PAPP-A/proMBP structure in lines 302-304:

“A recent cryo-EM structure visualized that both PAPP-A monomers cooperate to bind to two proMBP monomers (Zhong et al. 2022) in agreement with prior data obtained from biochemical experiments (Overgaard et al. 2003; Glerup et al. 2005).”

In lines 306-308, we cited the PAPP-A/STC2 structures as follows:

“The structural basis of PAPP-A inhibition by STC2 was also recently reported by two groups and indicated that a dimer of STC2 is bound by cooperation between LNR1, LNR2, and LNR3 regions and M1 domain of each monomer of PAPP-A (Kobbero et al. 2022; Zhong et al. 2022).”

Finally, it is unclear from this comment whether the reviewer thought that mentioning our previous PAPP-A/IGFBP5 structure was irrelevant to the manuscript as a whole or if the reviewer was simply suggesting that it was not relevant to emphasize our previous structure because other PAPP-A structures have been reported. We feel that we have now provided sufficient mention of other PAPP-A structures to avoid overemphasis of our prior publication. We also

respectively note that our previous publication containing the PAPP-A/IGFBP5 anchor peptide cryo-EM structure is relevant to this work because comparing IGFBP5 cleavage between PAPP-A and PAPP-A2 is a major focus of this manuscript, and our prior report is the only publication that has reported the PAPP-A/IGFBP5 anchor peptide structure.

-please be precise about what parts of the structure is reported (not just N-terminal part). Also not e.g. LNR1.

We are sorry if our wording caused confusion. To add additional clarity, in our abstract we explicitly state (lines 17-19):

“In this report, we present the single particle cryo-EM structure of the monomeric, N-terminal LG, MP, and the M1 domains (with the exception of LNR1 and LNR2 regions) of PAPP-A2 to 3.13 Å resolution.”

Please note, that this information was already included in our initial manuscript in the results section where we stated (lines 93-95):

“The structure was reconstructed to 3.13 Å resolution (Fig. 1b, Table S1, and Fig. S2a-S2d), which allowed us to build a model corresponding to residues 25-926, except for residues 373 – 420 in the LNR1 and LNR2 regions that featured ambiguous map density”.

By writing “the final structure contains the N-terminal LG, MP, and the M1 domains of PAPP-A2” (lines 95-96) we were simply clarifying that these residues correspond to the N-terminal region of PAPP-A2.

Introduction

-number of residues in mature protein is incorrectly specified as 1524??

We thank the reviewer for pointing this out and the number of amino acids for PAPP-A has been corrected (lines 36-38).

-sentence “Both PAPP-A2 and PAPP-A contain an N-terminal Laminin-G (LG) domain and a catalytic metalloprotease (MP) domain that itself contains two Lin12-Notch repeats (LNR1 and LNR2) (Boldt et al., 2006)”: Why is reference made only to Boldt 2006?? Boldt 2004 should be placed here also. Place references accurately, which appears to be journal style. MANY similar inaccuracies throughout the ms.

We apologize for this omission, and it has been corrected (lines 39-41):

“Both PAPP-A2 and PAPP-A contain an N-terminal Laminin-G (LG) domain and a catalytic metalloprotease (MP) domain that itself contains two Lin12-Notch repeats (LNR1 and LNR2) (Boldt et al. 2006; Boldt et al. 2004).”

-sentence: “Unlike PAPP-A that forms a trans-homodimer, PAPP-A2 exists as a monomer (Judge et al., 2022; Overgaard et al., 2001).” Missing citation = Weyer 2007. Weyer 2007 is the first demonstration that PAPP-A dimerizes in trans. Later both Judge 2022 and Kobberø 2022 confirmed this. Again, place references accurately to allow the reader to go back in the literature.

We apologize for not citing Weyer 2007 and we have rewritten this sentence to include citation to the original dimerization data as follows (lines 48-50):

“PAPP-A was first experimentally shown to form a *trans*-homodimer (Weyer et al. 2007) and was later visualized as a dimer by cryo-EM reconstruction in the apo form and when bound to IGFBP5 (Judge et al. 2022), and when bound to inhibitory proteins STC2 and proMBP (Kobbero et al. 2022; Zhong et al. 2022).”

-that PAPP-A2 exists as a monomer is NOT experimentally supported. Discovery paper (Overgaard 2001) shows that PAPP-A2 subunits DO NOT form a COVALENT SS-based dimer. However, Weyer 2007 shows that IN NATIVE PAGE, PAPP-A and PAPP-A2 migrates similarly, thus PAPP-A2 migrates as a dimer.

We thank the reviewer for pointing out this issue. We have revised our introduction to include citations to previous work regarding PAPP-A2 oligomerization. We now state (lines 50-54):

“Unlike PAPP-A, PAPP-A2 was originally shown to not form a covalent dimer by non-reducing PAGE (Overgaard et al. 2001), yet the majority of PAPP-A2 was shown to run at a similar migration as dimeric PAPP-A on native PAGE (Weyer et al. 2007). Purified recombinant PAPP-A2, however, was shown to exist as a monomer in solution as analyzed by size exclusion chromatography multi-angle light scattering (SEC-MALS) (Judge et al. 2022).”

SEC of the current study does not provide any info about size in this regard. Of importance, this has implications throughout the manuscript, including reasoning MANY places. Serious revision is required.

We now provide a SEC result of PAPP-A2 that includes molecular weight standards and PAPP-A for comparison (Fig. S3b) that shows that PAPP-A2 runs at a position consistent with a monomer. Please note that PAPP-A2 was also previously determined to have a molecular weight consistent with a monomer by SEC-MALS (Fig. S2d from Judge et al., 2022). We have now cited this in our manuscript (please see the response to the previous comment) and the prior result is also shown below:

PAPP-A2 SEC-MALS result from Figure S2d of Judge et al., 2022:

-also, must be discussed in relation to how refinement mask was define. Did the refinement mask cover a sub-volume only? Not clear at all.

We thank the reviewer for allowing us to clarify this point. Please see the image below:

The mask (gray surface) applied during 3D refinement covers the entire density of the map (enclosed yellow density). One molecule of PAPP-A fits the final map density very well as shown in Fig. 1b. Thus, no additional density exists that could represent a PAPP-A2 dimer. In our revision, we included the following sentence to better clarify how the mask was used (lines 703-704):

“The mask enclosed the entirety of the map, and no additional density was present that could account for a second copy of PAPP-A2.”

We have also included the image shown above in our revised manuscript as Fig. S2b to add clarity to our analysis.

-relating to the two previous points: 2D classification figure of Figure 2S suggests symmetry??

We thank the reviewer for this question, and we illustrate here why PAPP-A2 does not show symmetry. Below is a high resolution, zoomed in view of a portion of Fig. S2a that shows 2D class averages of PAPP-A2:

Some of the particles may *look* symmetric, but in fact they are not. If PAPP-A2 were symmetric this would imply that there was an identical copy (or set of domains) in the protein. PAPP-A2 is monomeric by cryo-EM analysis and functional experiments as discussed in the prior and following comments. The 2D class averages in the figure are not symmetric and symmetry was not used when generating the 3D map.

For comparison, please see below several images of PAPP-A complexes that clearly show two copies of PAPP-A and symmetry.

PAPP-A/STC2 complex in Fig. S1b from Kobbero et al. 2022:

b (Representative 2D classes from combined Dataset I and Dataset II)

PAPP-A/proMBP complex from Fig. S1h from Zhong et al 2022:

PAPP-A/IGFBP5 complex from Fig. 1e from Judge et al 2022:

e

These images help to further illustrate that while these PAPP-A complexes are dimers, PAPP-A2 is not.

-sentence “Also, PAPP-A binds to cell surface glycosaminoglycans (GAGs) via its CCP4 and CCP5 domains”: Is NOT correct. PAPP-A binds GAG via CCP3 and CCP4 (Laursen 2002b), not CCP4 and CCP5 as stated.

We thank the reviewer for pointing out this mistake and it has been corrected (lines 46-48):

“Also, PAPP-A binds to cell surface glycosaminoglycans (GAGs) via its CCP3 and CCP4 domains, but PAPP-A2 does not (Laursen et al. 2002).”

Based on the cited paper it cannot be concluded that “PAPP-A2 mainly exists in circulation”. For a discussion of that, see e.g. PMID 36718521.

We thank the reviewer for this helpful comment and have revised as stated in the response to the previous comment (lines 46-48):

-sentence: Our previous report together with other studies suggested various key determinants of PAPP-A required for cleavage of IGFBP4 but not for IGFBP5: 1) LNR center formation, 2) an interaction between the LG and CCP2 domains, and 3) trans-dimerization (Boldt et al., 2004; Judge et al., 2022; Weyer et al., 2007). It is very misleading to use this kind of wording. Only the LG-CCP2 point refers to “Our previous report” (= Judge 2022), and it is unclear what is meant by that – to what extent this is experimentally supported. Reference for point 1 is Boldt 2004. Reference for point 3 is Weyer 2007.

We agree that this section required revision and apologize if our previous wording was confusing. We have revised this sentence as follows to add clarity and attribute citation properly (lines 62-65):

“There are several key determinants of PAPP-A required for cleavage of IGFBP4 but not for IGFBP5 including 1) LNR center formation (Boldt et al. 2004), 2) an interaction between the LG and CCP2 domains (Judge et al. 2022), and 3) *trans*-dimerization (Boldt et al. 2004; Judge et al. 2022; Weyer et al. 2007).”

-sentence: “Interestingly, the A799V point mutation site is far from the predicted proteolytic site of PAPP-A2, and its inactivation mechanism is unclear.”: It should be stated specifically here what was found in the Dauber paper: That, A799V has no catalytic activity towards IGFBP3 and 5.

We thank the reviewer for this comment. We have revised this section to make the known effect of the A799V mutation clearer and also to better convey what is still unknown regarding this mutation as follows (lines 74-79):

“Frameshift, nonsense, and the A1033V point mutation in PAPP-A2 (referred to as “A799V” hereafter based on the mature sequence numbering) were shown to result in short stature in patients and abolish cleavage of IGFBP3 and IGFBP5 by PAPP-A2 (Babiker et al. 2021; Dauber et al. 2016). Interestingly, the A799V point mutation site is far from the proteolytic site of PAPP-A2 so the mechanism of how it abolishes PAPP-A2 proteolytic activity is unclear, but the mutant protein was previously reported to have lower expression compared to wildtype protein and was partially cleaved (Dauber et al. 2016).”

That the recombinant protein expresses much less efficiently compared to wild-type PAPP-A. AND that it is susceptible to autocleavage (that might be inactivating). Completely left out??

We thank the reviewer for pointing this out and in our introduction, we now include the following sentence (lines 78-79):

“...the mutant protein was previously reported to have lower expression compared to wildtype protein and was partially cleaved (Dauber et al. 2016).”

We also discuss the result of Dauber et. al. 2016 in our results section as mentioned in an earlier response (lines 133-136).

Results

-active site inactivated PAPP-A2 has previously been generated and analyzed (discovery paper = Overgaard 2001). It is incorrect to say “in agreement with analogous results with PAPP-A”. Reference to actual PAPP-A2 paper is required.

We thank the reviewer for the suggestion. We were aware of this reference and did cite it in the initial version of our manuscript when discussing the cryo-EM structure (lines 90-91):

“We used the reported catalytically inactivating E500Q mutant (Overgaard *et al.*, 2001) to prevent substrate cleavage, and reconstituted PAPP-A2 with IGFBP5 in attempts to resolve a substrate-bound complex.”

We also additionally reworded this sentence as explained in the response to the comment below.

-Figure 1e+f: Active site-inactivated PAPP-A2 previously shown (Overgaard 2001). Here a main figure?

Our purpose here was to use E500Q as a control for comparison for the other mutations that had not been previously reported, but we see how this may have been confusing. We reworded this sentence as below and cited Overgaard *et al.* 2001 (lines 116-119):

“Compared with wild type (WT) PAPP-A2, the H499A and E516A mutants abolish cleavage for both IGFBP5 and IGFBP3 like the previously reported E500Q mutant (Overgaard *et al.* 2001), in agreement with analogous results with PAPP-A (Boldt *et al.* 2001; Gaidamauskas *et al.* 2013) (Fig. 1e, Fig. S4a, and Fig. S4b; and Fig. 1f, Fig. S4a, and Fig. S4c, respectively).”

-Figure 3d: K128D shown previously. Here a main figure?

We thank the reviewer for this suggestion and the figure has been moved to Fig. S8a.

-sentence: “Therefore, the chain-like effect provides a straightforward explanation for the lack of activity in the PAPP-A2 patient mutation, while we note that we cannot completely rule out other effects of the A799V mutation on protein folding.”
: As mentioned above, compromised recombinant expression as reported in Dauber 2016 should be referred to also here.

Further to mention: Dauber reported intramolecular cleavage of the mutant, which may also cause disturbed structure/activity.

We thank the reviewer for this comment. As discussed in a response to a previous comment, we have removed the phrases “chain-like effect” and “straightforward explanation” from our revision and also included nano DSF data.

We have cited the previous observations by Dauber et. al. 2016 regarding the patient mutation in our revised manuscript in both the Introduction and Results as discussed in responses to previous comments. We also now include additional comparisons between our protein and the results from Dauber et al 2016 in (lines 138-140):

“While we did observe lower expression of PAPP-A2 A799V compared to wildtype protein, we did not observe significant truncation of PAPP-A2 A799V as protein quality was comparable to wildtype protein (Fig. S6a, Fig. S3a, and Fig. S3c).”

Therefore, it is critical to show the A799V variant under BOTH reducing and nonreducing conditions.

We thank the reviewer for this suggestion. We now show wildtype and A799V PAPP-A2 under both reducing and non-reducing conditions in Fig. S3a and also included the description of this result (lines 101-103):

“We confirmed the monomeric nature of recombinant PAPP-A2 by reducing and non-reducing SDS-PAGE (Fig. S3a) indicating that PAPP-A2 does not form a covalent dimer and in agreement with prior results (Overgaard et al. 2001).”

-sentence p7: “..and removal of the C-terminal domains of PAPP-A also does not hinder IGFBP5 cleavage (Judge et al., 2022).” This is a key finding about PAPP-A. Made in 2004 (Boldt 2004). Elaborated on in several following papers, most recently in Kobberø 2022. Citing Judge is narrow-minded and with no respect towards the scientific literature.

We sincerely apologize for this omission as we have changed citation in our revision as follows (lines 209-213):

“Removal of the C-terminus of the protein starting with the CCP1 domain (PAPP-A2₁₁₅₉) resulted in a 2.7- fold reduction in activity (Fig. 4c, Fig. S12a, and Fig. S12b), suggesting that this region could contribute to, but is not essential for IGFBP5 cleavage. This is not surprising as we do not observe density for these regions in the PAPP-A2 cryo-EM structure, the ML-PAPP-A2/IGFBP5 model does not predict their direct interactions with the anchor peptide, and removal of the C-terminal domains of PAPP-A also does not hinder IGFBP5 cleavage (Boldt et al. 2004; Judge et al. 2022).”

-indicate clearly for all PAPP-A2 SDS-PAGE gels whether the gels were run under reducing or nonreducing conditions.

We now make this clarification in the legend to Fig. S4a (lines 466-467):

“Please note that the gels shown here and all other gels shown in this report, with the exception of Fig. S3c and certain lanes in Fig. S3a, were run under reducing conditions.”

-a lot of excessive speculation in Results, which does not belong there. Also, in general it is often difficult to follow exactly what the authors are suggesting as experimental support.

We regret if our Results section was confusing. We have revised our wording significantly to make it clear to the reader which results are derived from experiments. We now more clearly specify when results were derived from ML models or MD simulations.

-may not be clear to the reader that some results are predictions based on predictions...

We thank the reviewer for the suggestion. We now state that our MD simulations are based on our ML model. We also state the limitations of modeling work in the Discussion section (lines 287-290):

“It is worth noting that the MD simulation was performed using the ML-PAPP-A2-IGFBP5 model as template, and we validated hypotheses derived from the model with functional data. We note that there are limitations of this mechanism from not having a starting experimental model and that future experimental structural studies would be helpful to fully validate our model.”

Discussion:

-Sentence: “Sequestered IGFs are liberated by cleavage of IGFBP proteins by PAPP-A2, PAPP-A, and other proteases”: Which other proteinases? In vitro, in vivo? Evidence? See e.g. PMID 36718521.

We agree that this was not the correct word choice for this sentence and have removed mention of “other proteases” here.

We should have been more specific. IGFBP3 has been shown to be cleaved by both ADAM12 and thrombin *in vitro* and we include this information in the discussion and our tone is *very mild* regarding the significance of this data (lines 299-300):

“IGFBP3 has been shown to be cleaved by both ADAM12 and thrombin *in vitro* (Shi et al. 2000; Kim et al. 2022), possibly suggesting alternative modes of processing.”

p9, “A recent structure revealed that both PAPP-A monomers cooperate to bind to two proMBP monomers (Zhong et al., 2022).” Wrong citation. Not revealed by a recent study. Stoichiometry/and binding was determined/analyzed in Overgaard 2003 (PMID 12421832) and Glerup 2005 (PMID 15647258).

We apologize as it was not our intent to omit mentioning prior studies. Our intention was to point out that PAPP-A binding to proMBP had recently been *visualized* by cryo-EM. We have revised this sentence as follows (lines 302-304):

“A recent cryo-EM structure visualized that both PAPP-A monomers cooperate to bind to two proMBP monomers (Zhong et al. 2022) in agreement with prior data obtained from biochemical experiments (Overgaard et al. 2003; Glerup et al. 2005).”

-sentence top of p9 “...the substrate recognition residues”: Not at all defined to a level that justifies reference like that.

We thank the reviewer for allowing us to correct this issue. We no longer use the term “substrate recognition residues” and instead use the term “predictions” when referring to the ML model. We have revised our discussion significantly regarding the ML model as follows (lines 277-279):

“We used machine learning to predict interactions between PAPP-A2 and IGFBP5 (Fig. 3a, Fig. 3b, and Fig. 3e) and combined with our cryo-EM structure, we used mutagenesis with cleavage assays to support these predictions by showing that PAPP-A2 is more vulnerable to mutation than PAPP-A (Fig. 3d and Fig. 3g).”

Potential over-interpretation of effects of mutation.

We thank the reviewer for this comment and have made significant revisions to avoid over-interpretation. In cases where structural-functional interpretations are made based on the ML model and tested by functional assays, we now indicate

that these results *support* the model to avoid overemphasizing the effects of the mutations. For instance, please see the response to the previous comment (lines 277-279) where it is stated that

“We used mutagenesis with cleavage assays to support these predictions by showing that PAPP-A2 is more vulnerable to mutation than PAPP-A..”

As another example, please see how we have revised our wording for the patch mutation data to avoid overemphasis (lines 157-159):

“Thus, the patch regions are sensitive to mutation and their long-range connection to the active site and residue A799 suggests a plausible mechanism of PAPP-A2 inactivation by the A799V patient mutation.”

Methods:

Cleavage assay: What happened after the gels were run? Staining? Blotting? Detection? Details lacking. PBS not defined here (or elsewhere? Calcium concentration???)

We thank the reviewer for allowing us to expand on our methods. The methods have been revised to include more experimental details. Gels were stained as now indicated (lines 621-624):

“The quenched reactions were applied to Bolt 4-12% Bis-Tris SDS-PAGE gels (Invitrogen, product #NW04122Box) run under reducing conditions in 1 X MES buffer (diluted from 20X stock, Invitrogen, product #B000202) and stained with Instant Blue Coomassie Protein Stain (abcam product #ab119211).”

The 1XPBS buffer is now properly defined and a product number is included in lines 602-603:

“1XPBS (154 mM NaCl, 1.06 mM KH₂PO₄, Na₂HPO₄, Corning product #21-040-CM)”

p17, relating to points about monomer/dimer: “PAPP-A/PAPP-A2 Proteins were concentrated and injected into a Superose 6 Increase 10/300 column (Cytiva) run in 1XPBS. PAPP-A2 eluted as a monomer while PAPP-A eluted as a dimer.”: This CANNOT be concluded. There are no figures showing co-runs, and the column was not even calibrated. Statement is based on assumption only. Again, native PAGE shows differently as mentioned above.

We thank the reviewer for this comment, and we have now provided additional experimental data to support our claim that PAPP-A2 is a monomer. We have addressed the gel filtration issue by including molecular weight standards in our gel filtration run (Fig. S3b) and referenced our prior SEC-MALS result from Judge et. al. (as discussed in response to a previous comment). We also ran wildtype PAPP-A2 and A799V mutant on native PAGE (Fig. S3c). Purified PAPP-A2 proteins run predominantly as a monomer in both experiments.

We now acknowledge procedural differences between our PAPP-A2 oligomerization analysis and previous results as follows (lines 105-108):

“Native PAGE analysis of our recombinant PAPP-A2 also indicated that it was predominantly monomeric (Fig. S3c), in contrast to a prior result obtained from Western blotting of PAPP-A2 overexpression cell culture media (Weyer et al. 2007).”

We have now holistically characterized the oligomerization state of PAPP-A2 by cryo-EM, non-reducing and reducing PAGE, native PAGE, and gel filtration and all results indicate that purified, recombinant PAPP-A2 is predominantly monomeric.

REVIEWERS' COMMENTS:

Reviewer #1 (Remarks to the Author):

The reviewers' suggested corrections have been made and the manuscript is improved through being more conservative with the conclusions with respect to the mechanism of effect of the mutation as well as the claims made from the ML data. I have no further comments/suggested corrections.

Reviewer #2 (Remarks to the Author):

Authors have made a reasonable attempt to address concerns raised.

Reviewer #3 (Remarks to the Author):

Concerns addressed in an ok manner.

Reviewer #1 (Remarks to the Author):

The reviewers' suggested corrections have been made and the manuscript is improved through being more conservative with the conclusions with respect to the mechanism of effect of the mutation as well as the claims made from the ML data. I have no further comments/suggested corrections.

We thank the reviewer for the encouraging comments!

Reviewer #2 (Remarks to the Author):

Authors have made a reasonable attempt to address concerns raised.

We thank the reviewer for the encouraging words!

Reviewer #3 (Remarks to the Author):

Concerns addressed in an ok manner.

We thank the reviewer for the encouraging words!